**Best practices for precipitation sample storage for offline studies of ice nucleation in marine and coastal environments**
Charlotte M. Beall[1], Dolan Lucero[2], Thomas C. Hill[3], Paul J. DeMott[3], M. Dale Stokes[1], Kimberly A. Prather[1,4]
[1]Scripps Institution of Oceanography, University of California San Diego, La Jolla, CA, 92037, USA
[2]Department of Earth and Environmental Science, New Mexico Institute of Mining and Technology, Socorro, NM, 87801,
USA
[3]Department of Atmospheric Sciences, Colorado State University, Fort Collins, CO, 80523, USA
[4]Department of Chemistry and Biochemistry, University of California San Diego, La Jolla, CA, 92093, USA
*Correspondence to*: Kimberly A. Prather (kprather@ucsd.edu)
**Abstract.** Ice nucleating particles (INPs) are efficiently removed from clouds through precipitation, a convenience of nature
for the study of these very rare particles that influence multiple climate-relevant cloud properties including ice crystal
concentrations, size distributions, and phase-partitioning processes. INPs suspended in precipitation can be used to estimate
in-cloud INP concentrations and to infer their original composition. Offline droplet assays are commonly used to measure INP
concentrations in precipitation samples. Heat and filtration "treatments" are also used to probe INP composition and size
ranges. Many previous studies report storing samples prior to INP analyses, but little is known about the effects of storage on
INP concentration or their sensitivity to treatments. Here, through a study of 15 precipitation samples collected at a coastal
location in La Jolla, CA, USA, we found INP concentration changes up to > 1 order of magnitude caused by storage to
concentrations of INPs with warm to moderate freezing temperatures (-7 to -19 $^{\circ}$C). We compared four conditions: 1.) storage
at room temperature (+21-23 $^{\circ}$C), 2.) storage at +4 $^{\circ}$C 3.) storage at -20 $^{\circ}$C, and 4.) flash freezing samples with liquid nitrogen
prior to storage at -20 $^{\circ}$C. Results demonstrate that storage can lead to both enhancements and losses of greater than one order
of magnitude, with non-heat-labile INPs being generally less sensitive to storage regime, but significant losses of INPs smaller
than 0.45 μm in all tested storage protocols. Correlations between total storage time (1-166 days) and changes in INP
concentrations were weak across sampling protocols, with the exception of INPs with freezing temperatures >= -9 $^{\circ}$C in
samples stored at room temperature. We provide the following recommendations for preservation of precipitation samples
from coastal or marine environments intended for INP analysis: that samples be stored at -20 $^{\circ}$C to minimize storage artifacts,
that changes due to storage are likely an additional uncertainty in INP concentrations, and that filtration treatments be applied
only to fresh samples. At the freezing temperature -11 $^{\circ}$C, average INP concentration losses of 51%, 74%, 16% and 41% were
observed for untreated samples stored using the room temperature, +4 $^{\circ}$C, -20 $^{\circ}$C, and flash frozen protocols, respectively.
Finally, the estimated uncertainties associated with the 4 storage protocols are provided for untreated, heat-treated and filtered
samples for INPs between -9 and -17 $^{\circ}$C.

## 1. Introduction

In-cloud ice crystals and their formation processes are critical features of Earth's radiative and hydrological balance, affecting multiple climate-relevant cloud properties including cloud lifetime, reflectivity, and precipitation efficiency (DeMott et al., 2010; Lohmann, 2002; Lohmann and Feichter, 2005; Tan et al., 2016; Creamean et al., 2013). Ice nucleating particles (INPs) impact ice crystal concentrations and size distributions in clouds by triggering the freezing of droplets at temperatures above the homogeneous freezing point of water ( -38 °C).

INPs have been sampled in clouds and precipitation for decades (e.g. Rogers et al., 1998; Vali, 1971; Vali, 1966) to measure abundances, probe their compositions and investigate the extent to which they impact the properties of clouds. There are several caveats to consider when inferring in-cloud INP concentrations or properties from precipitation samples (Petters and Wright, 2015), including "sweep-out" of additional INPs as the hydrometeor traverses the atmosphere below the cloud (Vali, 1974) and heterogeneous chemistry due to adsorption or absorption of gases (Hegg and Hobbs, 1982; Kulmala et al., 1997; Lim et al., 2010). However, assessing the composition of INPs in precipitation samples is more straightforward than cloud particles. Thus, the number of publications reporting measurements of INP concentrations in precipitation has increased over the past decade. Numerable insights have been obtained in previous precipitated-based INP studies, including the efficient depletion of INPs relative to other aerosols of similar size in precipitating clouds (Stopelli et al., 2015), constraints on minimum enhancement factors for secondary ice formation processes (Petters and Wright, 2015), and the identification, characteristics and distribution of various INP populations (e.g. Christner et al., 2008a; Hader et al., 2014; Stopelli et al., 2017). INP concentrations in precipitation have been used to estimate in-cloud concentrations, based on assumptions that the majority of particles (86%) in precipitation originate from the cloud rather than the atmospheric column through which the hydrometeor descended (Wright et al., 2014). Along the same line of reasoning, INPs in precipitation have also been used to infer sources and composition of in-cloud INP populations (e.g. Martin et al., 2019 and Michaud et al., 2014, respectively).

A number of online (real-time) and offline (processed post-collection) techniques exist for measurement of INPs for each ice nucleation mechanism, including condensation, deposition, immersion and contact freezing. However, as some simulations have shown that immersion mode freezing is the dominant mode of primary freezing in the atmosphere between 1000 and 200 hPa (Hoose et al, 2010), most techniques target immersion freezing. Despite the lack of time resolution, offline techniques enable measurement of INPs at modest supercooling (e.g. up to -5 °C) and temperature regimes where concentrations typically fall below detection limits of online instruments (DeMott et al., 2017). Offline instruments capable of immersion mode INP measurement include a number of droplet assays, in which sample suspensions are distributed among an array of droplets that are then cooled and frozen (e.g. Budke and Koop, 2015, Harrison et al., 2018, Hill et al., 2014, Whale et al., 2015) as well as other systems in which water is condensed onto particles collected on substrates prior to cooling and freezing (e.g. Mason et al., 2015). As they are designed for analysis of liquid suspensions, droplet freezing assay techniques are commonly used for measurement of INPs suspended in precipitation (e.g. Creamean et al., 2019, Rangel-Alvarado et al., 2015, Michaud et al., 2015, Stopelli et al., 2014, Wright et al., 2014).

Many studies report results from samples stored prior to processing. Storage protocols vary widely, including total storage
time, time between collection and storage, and temperature fluctuations between collection, shipment and storage (if these
details are provided at all, see summary Table S1).  Storage temperatures range from -80 $^{\circ}$C (Vali et al., 1971) to +4 $^{\circ}$C (e.g.
Petters and Wright, 2015, Failor et al., 2017, Joyce et al., 2019), yet generally samples are stored between +4 $^{\circ}$C and -20 $^{\circ}$C.
Reported storage intervals range between hours (Schnell et al., 1977; Christner et al., 2008) to 48 years (Vasebi et al., 2019).
The understanding of storage effects on INPs suspended in precipitation is limited (Petters and Wright, 2015), and the
understanding of storage effects on INPs collected on filters is similarly lacking (Wex et al., 2019). Stopelli et al. (2014a)
studied INP concentrations in a snow sample stored at +4 $^{\circ}$C and observed a decrease in the concentration of INPs active at -
10 $^{\circ}$C over 30 days by a factor of ~2. Schnell (1977) reported significant losses in fog and seawater samples after storage at
room temperature for short periods (6-11 hours).  Several studies have reported on the lability of commercially available dust
and biological IN entities in storage above 0 $^{\circ}$C or under freezing conditions, including Arizona Test Dust and SnoMax®
(Perkins et al., 2020; Polen et al., 2016; Wex et al., 2015), and similar labilities could affect the INPs of similar composition
in precipitation samples (Creamean et al., 2013; Martin et al., 2019).  Considering the abundance of precipitation based INP
studies, the lack of bounds on potential impacts of storage on INP concentration measurements represents a critical
uncertainty in conclusions derived from data on stored samples. Furthermore, to determine INP activation mechanisms and
composition, previous studies have applied "treatments" to precipitation samples, including heat, filtration, enzymes and
peroxide, (e.g. Hill et al., 2014) but it is unknown to what extent storage affects the results of such experiments.
Here we investigate the effects of four storage protocols on INPs using 15 precipitation samples collected between 9/22/2016
and 11/22/2019 at two coastal sites at Scripps Institution of Oceanography, La Jolla, CA, USA: 1.) storage at room temperature
(+ 21-23 $^{\circ}$C) , 2.) storage at +4 $^{\circ}$C ("refrigerated"), 3.) storage at -20 $^{\circ}$C ("frozen"), and 4.) flash freezing samples with liquid
nitrogen prior to storage at -20 $^{\circ}$C ("flash frozen").  The abundance of previous studies that report storage between +4 $^{\circ}$C and
-20 $^{\circ}$C motivated the choice of techniques 2 and 3 (see Table S1). Room temperature storage was chosen to provide context
as a "worst-case scenario", and the flash freezing technique was chosen to investigate whether any changes of INP
concentrations could be mitigated by instantaneous freezing prior to storage.  The 15 precipitation samples in this study were
divided into several replicates so that the concentration of INPs could be measured in untreated, heated, and filtered samples
when fresh, and again after storage using the 4 techniques described above. Sample replicates were additionally processed at
2 different points in time to investigate the effects of total storage time on INP concentration measurements.  Enhancements
and losses of INPs according to storage protocol and treatment are reported, as well as recommendations for storage protocols
that best preserve INPs in untreated, heated, and filtered precipitation samples from marine or coastal environments.
**2. Methods**
**2.1 Precipitation Sample Collection**
Precipitation samples were collected at two coastal locations at Scripps Institution of Oceanography (32.87 N 177.25 W): the
rooftop of the Ellen Browning Scripps Memorial Pier laboratory (32.8662 °N, 117.2544 °W) (10 meters above sea level) and
the rooftop of a storage container next to Isaacs Hall (32.8698 °N, 117.2522 °W,  58 meters above sea level, 500 m inland).
Collection technique varied based on location. At the SIO pier, the Teledyne ISCO model 6712 commercial water sampler
(Teledyne ISCO, Inc., US) was used. A plastic funnel, 27 cm in diameter, and Tygon tubing, connected the sampler inlet to
the water sampler's distributor arm. The samples were distributed via the distributor arm into one of twenty-four 1-liter
polypropylene bottles on an hourly time interval. Bottles corresponding to consecutive 1-hour time intervals were combined
when the hourly precipitation volume was insufficient for sample separation and analysis (< 50 mL per bottle). At the Isaacs
Hall location, an ISO 6706 plastic graduated cylinder and plastic funnel, 27 cm in diameter, was used for precipitation
collection. At both sites, ring stands supported the collection funnels approximately 60 cm above the rooftop. All funnels,
tubing, cylinders, and bottles were cleaned with 10% hydrogen peroxide for 10 minutes and rinsed with milli-Q purified water
three times immediately before each sampling event.  Satellite composites from the National Weather Service Weather
Prediction Center's North American Surface Analysis Products were used for synoptic weather analysis to generally
characterize each rain event (see Table 1).  Atmospheric river (AR) events were identified using the AR Reanalysis Database
described in (Guan and Waliser, 2015) and (Guan et al., 2018).
**2.2 Storage Protocols**
The following sample storage protocols were used: frozen at -20 °C, refrigerated at 4 °C, room temperature (21 - 23 °C), and
flash freezing, or flashing with liquid nitrogen (-196 °C) before frozen at -20 °C.   All techniques except storage at room
temperature are commonly used for offline INP analysis (see Table S1). Excluding the samples that were flash frozen, all
samples were stored in 50 mL sterile plastic Falcon® tubes (Corning Life Sciences, Corning, NY, USA). Flash frozen samples
were stored in polypropylene 5 mL cryovials. Prior to storage, 25 - 50 mL bulk sample aliquots were distributed from collection
bottles into Falcon® tubes, shaking bottles ~10 s between each distribution. Not all samples were stored using all four of the
storage protocols due to limited volume for some samples. See Tables 2-4 for a summary of the number of samples studied for
each storage protocol. Precipitation samples were stored for varying intervals between 1 and 166 days to investigate effects of
storage time on INP concentrations.  INP measurements were made in two or three time steps: within two hours of collection,
and once or twice after storing using one of four storage protocols described above, depending on volume. Stored and fresh
samples were analysed in three treatment conditions: 1) raw untreated precipitation, 2) heated over a 95 °C water bath for 20
minutes and 3) filtered through a 0.45 µm surfactant-free cellulose acetate syringe-filter (Thermo Scientific™ Nalgene™,
Waltham, MA, USA). Heat treatments and filters were applied to samples just prior to processing (i.e. treatments were not
applied to samples prior to storage).
**2.3 INP Analysis**
The automated ice spectrometer (AIS) is an offline immersion-mode freezing assay which is described elsewhere (Beall et al.,
2017). Briefly, 50 uL aliquots of sample are pipetted into two sterile 96-well polypropylene PCR plates. The plates are inserted
into an aluminium block, machined to hold PCR plates, that sits in the coolant bath of a Fisher Scientific Isotemp® Circulator.
A thermistor placed atop the left side of the aluminium block, below the PCR plate, recorded temperature. An acrylic plate
separated the PCR plates from the ambient lab air. In the headspace between the acrylic plate and the PCR plates, nitrogen gas
flowed at a flow rate of 14 Lpm to reduce temperature stratification in the samples (Beall et al., 2017). The nitrogen gas was
cooled before emission by passing through the chiller via copper tubing. A 0.5 Megapixel monochrome camera (Point Grey
Blackfly 0.5MP Mono GigE POE) performed the image capture. Custom LabView software controlled the camera settings,
the rate the chiller cooled, and displayed the temperature of the thermistor.
A control milli-Q water sample is used, typically in the first 30 wells of each sample run, to detect contamination and for
subsequent INP concentration calculations. Thirty wells were used per sample to achieve a limit of detection of 0.678 IN mL
$^{-1}$. For each run, the chiller was cooled to -35°C. As the chiller cools the sample plates (1 ˚C/min), the custom LabView virtual
instrument records the location and temperature of the freezing event as they occur. Freezing events are detected by the change
in pixel intensity of the sample as it changes from liquid to solid.
**2.4 Particle Size Distributions**
Size distributions of insoluble particles suspended in the fresh and stored precipitation samples were measured using the Multi-
sizing Advanced Nanoparticle Tracking Analysis (MANTA) ViewSizer 3000 (Manta Instruments Inc.). The Manta ViewSizer
3000 applies multi-spectral particle tracking analysis (m-PTA) to obtain size distributions of particles of sizes between 10 and
2000 nm with three solid-state lasers with wavelengths of 450 nm, 520 nm and 650 nm. m-PTA has been shown to outperform
traditional dynamic light scattering (DLS) techniques when measuring polydisperse particles in suspension (McElfresh et al.,
2018). For analysis, 300 videos of the illuminated particles in suspension are recorded, each 10 seconds in length. The software
tracks each particle individually, obtaining particle size and number concentration from their Brownian motion and the imaged
sample volume.

**3 Results**
**3.1 INP concentrations in fresh precipitation samples**
Figure 1 shows INP concentrations of 15 coastal rain samples, collected in a variety of meteorological conditions including
scattered, low coastal rainclouds, frontal rain, and atmospheric river events (see Table 1). Observations generally fall within
bounds of previously reported INP concentrations from precipitation and cloud water samples (grey shaded region, adapted
from Petters and Wright, 2015). Observed freezing temperatures ranged from -4.0 to -18.4 ⁰C, with concentrations up to the
limit of testing at $10^5$ INP L$^{-1}$ precipitation. AIS measurement uncertainties are represented with 95% binomial sampling
intervals (Agresti and Coull, 1998).
Following the assumptions in (Wright and Petters, 2015) to estimate in-cloud INP concentrations from precipitation samples
(i.e. condensed water content of 0.4 g m$^{-3}$ air), observations of INP concentrations in fresh precipitation samples are
additionally compared to studies of field measurements conducted in marine and coastal environments. Figure 1 shows that
atmospheric INP concentration estimates compare with INP concentrations observed in a range of marine and coastal
environments, including the Caribbean, East Pacific, and Bering Sea, as well as laboratory-generated nascent sea spray aerosol
(DeMott et al., 2016). However, two of the warmest-freezing INP observations in Fig. 1 (at -4.0 and -4.75 °C) exceed
temperatures commonly observed in marine-influenced atmospheres, precipitation and cloudwater samples.
In 5 of the 15 heat-treated samples, INP concentrations were increased by 1.9 – 13X between -9 and -11 °C (see Discussion).
Excluding these 5 samples, the fraction of heat-resilient INPs varied between samples and generally increased with decreasing
temperature.  Geometric means and standard deviations of heat-treated:untreated INP ratios were 0.40 ×/÷ 1.9, 0.51 ×/÷ 2.0,
and  0.62 ×/÷ 2.1 at -11 , -13, and -15 °C respectively.
Fractions of INPs < 0. 45 µm also varied between samples, with geometric means and standard deviations of 0.48 ×/÷ 1.73,
0.30 ×/÷ 3.4 and 0.37 ×/÷ 1.9 at -11, -13, and -15 °C respectively. Mean values of heat-resilient INP fractions and INPs < 0.45
µm were calculated using the geometric mean, which is more appropriate than the arithmetic mean for describing a distribution
of ratios (Fleming and Wallace, 1986).
**3.2 Effects of sample storage on INP concentration measurements**
INP concentrations of stored replicate samples are compared with original fresh precipitation samples in Figures 2-4, calculated
in successive 2 °C increments between -7 and -19 °C.  This temperature range was chosen for the analysis because most fresh
precipitation samples exhibited freezing activity between -7 and -19 °C. Numbers of datapoints in Figs 2-4 differ across the
temperature intervals due to limits of detection (i.e. ratios were not calculated at temperatures where zero or all wells were
frozen in the fresh and/or stored sample).
All stored:fresh ratios were calculated from cumulative INP distributions in 2 °C intervals, meaning that the INP concentration
in each interval is inclusive of the concentration in all of the preceding (warmer) temperature intervals.  The choice of the
cumulative distribution was motivated by the fact that it is standard in INP studies to report INP concentrations in terms of the
cumulative distribution, and it is important to consider impacts of storage on cumulative INP distributions and any conclusions
derived from them. Thus, in this study, deviations observed in a stored sample are not necessarily independent, i.e. the
sensitivity of INPs to storage in one temperature interval could impact the observed changes in all of the following (colder)
temperature interval.  For example, in fresh untreated precipitation samples (see Fig. 1), 32% of the INP concentration
calculated at -11 °C activated in one of the preceding (warmer) 2 °C temperature intervals. At -17 °C, this fraction is increased
to 46%.
To investigate correlations between sample storage time and INP enhancements or losses, duplicate samples were archived
(when sufficient volume was available) so that each sample could be processed at two distinct points post-collection (see
example Fig. S1). For INPs with freezing temperatures >= -9 °C in samples stored at room temperature, time is moderately
correlated with changes in INP concentrations ($R^2 = 0.58$).  Figure S5 shows how losses of warm-freezing INPs in samples
stored at +4 °C and room temperature impact the cumulative INP spectra for a select sample.  Beyond these exceptions, little
to no correlation between storage time and INP enhancements or losses was found for untreated, heated and filtered samples
(see Figs S1-S4). This indicates that most of the changes in INPs observed may occur on shorter timescales than those studied
here, i.e. < 24 hours.
Figure 2 shows the ratio of stored sample to fresh sample INP concentrations for untreated precipitation samples stored under
four conditions: **(a)** room temperature (21 – 23 °C), **(b)** refrigerated (+ 4 °C), **(c)** frozen (-20 °C) and **(d)** flash frozen with
liquid nitrogen before storing at -20 °C.  Markers above the 1:1 line indicate enhancements in INP concentration from the fresh

sample, while markers below indicate losses. For each temperature interval containing data from at least two sets of replicate samples, the average difference in stored:fresh concentration ratios between replicates are represented with grey bars to indicate measurement variability. Replicate samples were processed for each storage protocol so that impacts of sample handling can be distinguished from storage impacts. For example, if settling occurs in bulk rain samples that are then divided into smaller volumes prior to storage, INP concentrations may differ between replicates of the bulk sample. Thus, it is assumed that INP concentration changes that are greater than differences between replicates (grey bars in Figs 2-4) can be attributed to storage impacts. We also assume that stored:fresh INP concentration ratios of 1:1 indicate insensitivity to storage, although it is possible that enhancements and losses of equal magnitude could also result in a 1:1 concentration ratio.

Finally, Fisher's Exact Test was applied to frozen and unfrozen well fractions between each stored sample and its corresponding fresh sample at each of the 2 $^\circ$C temperature intervals. Stored sample frozen well fractions that were significantly different ($p < 0.01$) from fresh sample frozen well fractions are indicated with filled markers. The term "significant" henceforth is intended to describe INP losses or enhancements that correspond to frozen well fractions that are determined to be significantly different from corresponding fresh sample frozen well fractions, according to Fisher's Exact Test (i.e. filled markers in Figs. 2-4). Results in Fig. 2 show that significant enhancements or losses of INPs occurred in all storage protocols between -9 and -17 $^\circ$C, and that on average, stored samples exhibit INP losses (as indicated by the mean change in each temperature interval). In frozen and flash frozen samples, all enhancements and losses fall within ± 1 order of magnitude, whereas several significant INP losses beyond 1 order of magnitude are shown in room and refrigerated samples. INP concentration changes >= 1 order of magnitude are greater than changes in the ratios of the total insoluble particle population 10 – 2000 nm during storage (see Fig. S6). This indicates that the INPs in these samples are more sensitive to storage than the total insoluble particle population. Fig. S5 illustrates the impacts of the 4 storage protocols on the full IN spectra of a select untreated precipitation sample at two time intervals, 27 days and 64 days after collection.

Figure 3 shows the effects of storage on INP observations in heat-treated precipitation samples. Non-heat-labile INPs represented the majority (62% on average at -15 $^\circ$C, see Sec. 3.1) of the total INPs observed in the fresh samples (i.e. 38% of the INPs in fresh samples were heat-labile). Fewer significant losses of non-heat-labile INPs are observed for heat-treated samples stored at room temperature and at 4 $^\circ$C compared with untreated samples. Again, slightly fewer (2-3) of the total frozen and flash frozen samples exhibit significant losses and enhancements. All observations other than the one significantly enhanced sample in (b) fall within ranges of stored:fresh ratios observed in the total insoluble particle population (see Fig. S7, within an order of magnitude). This demonstrates that non-heat-labile INPs are generally less sensitive to storage than the total INP population (Fig. 2).

Effects of storage protocol on INP concentrations of filtered precipitation samples are shown in Figure 4 (0.45 μm syringe filter, see Sect. 2.2 for details). INPs > 0.45μm represented the majority (52 and 63 % on average at -11 and -15 $^\circ$C, respectively, see Sec. 3.1) of total INPs measured in the fresh precipitation samples. A higher number of filter-treated samples exhibit significant losses across all 4 storage types when compared with the untreated samples. Furthermore,

significant losses > 1 order of magnitude are observed across all storage types indicating that INPs < 0.45 μm are generally
more sensitive to storage than the total INP population present in precipitation samples.
As the stored:fresh ratios follow a log-normal distribution (one-sample Kolmorgorov-Smirnov test), the uncertainties
associated with storage and 95% confidence intervals were calculated in using the geometric mean and standard deviation of
ratios of unique samples only between -9 and -17 °C (i.e. omitting any replicates, see Tables 5-7). . .

**4. Discussion**
The challenge in selecting a storage protocol for atmospheric samples (e.g. precipitation, cloud water, ambient atmosphere) is
that the INP population composition is unknown, diverse, and the impact of any given technique on the different species may
vary. Many types of aerosols can serve as INPs, including dusts, metals and metal oxides, organic and glassy aerosols,
bioaerosols, organic and mineral soil dust, and combustion products (Kanji et al., 2017). The aim of this study was to identify
a storage protocol that best preserves the concentrations and characteristics of the general INP population observed in
precipitation samples collected in a coastal environment. To this end, the impacts of 4 storage protocols on 15 untreated,
heated, and filtered precipitation samples collected between September 22, 2015 and November 22, 2019 in La Jolla, CA were
investigated by comparing measured INP concentrations between fresh and stored replicates. The fractions of INPs > 0.45 μm
observed in this study  varied between 52 and 63% at -11 and -15 °C,  respectively.  Excluding the five heat-treated samples
in which INP concentrations were enhanced (e.g. 1.9 - 13X between -9 and -11 °C), the average fraction of non-heat-labile
INPs varied between 40 and 62% at -11 and -15 °C, respectively.  INP enhancements in heat-treated samples are unexpected,
as heat-treatments are typically applied assuming that heat destroys proteinaceous (e.g. biological) INPs. The causes of INP
enhancements in heat-treated samples are unknown and have only been reported in coastal precipitation samples (Martin et
al., 2017) and nascent sea spray aerosol (McCluskey et al., 2018).  Possible sources include the redistribution of dissolved IN-
active molecules onto particles (McCluskey et al. 2018), and the release of IN-active content from cells (McCluskey et al.
2018, Wilson et al. 2015).  These findings demonstrate that in samples influenced by marine sources, a superposition of both
positive and negative ΔINP in samples could result in the observed changes in INP concentrations post heat-treatment.
Additionally, the INP freezing temperatures and concentrations observed in this study compare with INPs observed in studies
of marine and coastal environments (Fig. 1).  As spectra in this regime (-5 to -20 °C and $10^{-5}$ to $\sim 10^{-1}$ per L air, respectively)
cluster distinctly by source type (see Fig. 1-10 in Kanji et al., 2017), Fig. 1 indicates that the dominant sources to air masses
sampled in this study were marine. Considering that data in this study compare well with marine and coastal INPs from a
variety of marine-influenced air masses (DeMott et al., 2016, Yang et al., 2019), the findings herein are likely relevant to
samples from other marine and coastal environments.
While mean INP changes are within a factor of ~2 or less of fresh sample INP concentrations for all protocols except "Room
temperature" (Table 5), none of the 4 storage protocols prevented significant losses or enhancements of INP concentrations in
all samples (Fig. 2), indicating that INP concentration measurements on fresh precipitation are superior to measurements on
stored samples.  95% confidence intervals in Table 5 span losses > 1 order of magnitude in all protocols across multiple

temperature intervals. These uncertainties equal or exceed INP measurement uncertainties (1-2 orders of magnitude) at temperatures > -20 °C due to discrepancies between instruments (DeMott et al., 2017). If correspondence within 1 order of magnitude (or 2-3 °C) is desired, uncertainties associated with storage should also be considered in studies using samples from coastal or marine environments. Thus, uncertainty distributions provided in Tables 5-7 can be used to evaluate observed INP concentrations and responses to treatments in the context of potential changes due to storage. However, the degree to which INP sensitivity to storage varies by INP source (e.g. with soil-derived INP populations) remains to be tested.

Samples stored under freezing and flash freezing conditions exhibited fewer changes overall compared to refrigerated samples. For example, at the INP activation temperature of -13 °C, in the rain sample that exhibited the highest sensitivity to storage, over 20% of the original concentration was preserved in the frozen sample, whereas only 5% of the original concentration was preserved in the refrigerated sample. These losses are more extreme than those of (Stopelli et al., 2014b), which demonstrated that INP concentrations of a snow sample refrigerated over 30 days decreased only two-fold from 0.027 to 0.013 $L^{-1}$ at -10 °C. Despite the range of enhancements and losses of heat-sensitive INPs observed in fresh samples, non-heat-labile INPs were generally less sensitive to storage than the total INP population, and with the exception of samples stored at room temperature, all techniques yielded similar results with fewer enhancements or losses. Interestingly, INPs < 0.45 μm exhibited more sensitivity to all storage conditions tested than the total INP population, with significant losses (Fishers Exact Test, $p < 0.01$) observed in several samples leaving between 25% and 3% of the value observed in the original fresh sample. Losses of INPs < 0.45 μm in samples stored at room temperature and +4 °C were comparable to the losses of total INPs in untreated samples and are likely a result of chemical aging in solution. However, losses of INPs < 0.45 micron in samples stored at -20 °C (both frozen and flash frozen) exceeded losses observed in the corresponding untreated samples. This is surprising given that a large fraction of INPs in this study were resilient to heat treatments of +95 °C. Lacking the identities of INPs observed in this study, a clear mechanism for their losses remains elusive. However, we offer the following points for consideration. It is well known that as a solution freezes, some solute is incorporated into the crystal and some is rejected, leading to enrichment of the solution phase and aggregation of dissolved or colloidal organic matter (Butler, 2002). Thus, as precipitation samples are freezing, small organic INPs may be lost simply due to aggregation in channels of enriched solute. In coastal precipitation samples for example, INPs may be so "lost" as the increased salinity in solution-phase channels destabilizes small suspended particles, allowing them to coagulate and settle (Jackson and Burd, 1998). Another possibility is that as the solution phase is enriched during freezing, smaller INPs may be adsorbing onto the surface of larger particles. The size distributions of total insoluble particles in the frozen samples show that most samples exhibit losses between 0-500 nm after storage and enhancements in sizes > 500 nm (see Fig. S6). This effect is not observed for samples stored at room temperature or at +4 °C.

Changes in the total insoluble particle size distribution (± 1 order of magnitude between 10 and 2000 nm, see Figs S6 and S7) may also have contributed to the observed INP concentration enhancements. Potential mechanisms for INP enhancements include increases in the number concentration of small particles due to breakup of loosely clumped masses of smaller particles, the redistribution of dissolved IN-active molecules onto particles (McCluskey et al. 2018), and the release of IN-active content from cells (McCluskey et al. 2018, Wilson et al. 2015) during cell death and lysis post freezing (Mazur et al., 1984).

Previous studies on precipitation collected along the California coast have demonstrated the contribution of dust, marine and terrestrial bioparticles to INPs in precipitation (Levin et al., 2019; Martin et al., 2019). Considering that well-characterized IN-active dust and biological standards (Arizona Test Dust and Snomax®, respectively) are sensitive to storage conditions, it is possible that dust or biological INPs contributed to the observed INP changes. Perkins et al. (2020) found that the IN-ability of Arizona Test Dust is degraded in most conditions, including aging in deionized water for 1 day, and results from Polen et al. (2016) show that the most efficient (i.e. warmest freezing) components of biological ice nucleators are also the most labile and sensitive to storage.

The observed distributions of INP concentration changes in stored precipitation samples have implications for the interpretation of heat and filtration treatment experiments. As heat denatures proteins, heat treatments are commonly used to infer contributions of proteinaceous or cellular contributions to INP populations, and filters are commonly applied to identify observed INP size ranges (e.g. McCluskey et al., 2018). For example, a typical analysis involves a comparison of the INP spectrum of an untreated sample to that of the heat-treated or filtered sample, and information about the sizes and biological composition of INPs are derived from this comparison. Our results demonstrate that these treatments may yield different results if treatments are applied to stored samples. Any losses of INPs due to filtering or heat application could be confounded by significant enhancements or losses caused by storage (up to > 1 order of magnitude), resulting in inaccurate conclusions about INP characteristics. In this study, a large fraction (30% to 48%, on average) of INPs observed in fresh precipitation samples were < 0.45 μm. Considering this and that INPs < 0.45 μm exhibit significant losses across all storage types, there is a risk that filter-treatments on stored samples in this study would lead to the underestimation of INPs < 0.45 μm. Losses of heat-labile INPs in storage could also impact treatment outcomes on stored samples. Assuming negligible effects of storage on the heat-treated sample but losses due to storage in the untreated sample (e.g. as was shown to be most likely for untreated samples stored at +4 °C), INP spectra of heat-treated samples could appear to indicate the entire INP population was heat-insensitive. This effect was observed in several samples across storage types (see Fig. S8).

## 5. Conclusions

Based on all observations in this study, we provide the following recommendations for precipitation samples collected in coastal and marine environments for offline INP analyses:

1. Of the 4 storage protocols tested, none prevented changes in INP concentrations across all samples between -7 and -19 °C. However, whenever processing fresh samples is not possible, our results demonstrate that storage at -20 °C causes the least changes in INP concentrations.

2. Estimates of uncertainty attributed to storage impacts and 95% confidence intervals for INP measurements obtained from stored samples are provided (see Tables 5-7).

3. Flash freezing with liquid nitrogen before storing at -20 °C did not improve conservation of INPs.

4. With the exception of warm-freezing INPs (freezing temperatures >= -9 °C) in samples stored at room temperature, we found little to no correlation between changes in INP concentrations and storage intervals on timescales between

1-166 days, indicating that most enhancements or losses are likely happening during freezing or on timescales < 24
336        hours.
5.   INPs that are insensitive to heat treatments are also less sensitive to storage. However, potential enhancements or
338        losses due to storage (e.g. an average loss of 50% for INPs with freezing temperatures >= -15 °C in samples stored at
339        -20 °C) should be treated as additional uncertainty in measurements of INP concentration when comparing heat-
340        treated with untreated INP spectra.
6.   Due to the significant losses of INPs < 0.45 μm in storage, regardless of protocol, we recommend applying filtration
342        treatments to fresh samples exclusively.

As measurements of INPs suspended in precipitation samples are used to infer in-cloud INP composition and
concentration estimates, they represent important contributions to studies of links between aerosols, cloud processes and
precipitation outcomes. This study derives bounds and correction factors for the impacts of storage on INPs and treatment
outcomes from changes in INPs observed in coastal precipitation samples. However, it remains to be seen how INP
sensitivity to storage varies by environment or INP composition. Further studies are needed to bracket storage effects on
INP populations with various distributions of terrestrial and marine sources, as well as on heat-labile (biological) INPs,
and INPs with colder activation temperatures. These studies could additionally benefit from analysis on how storage
impacts differential INP spectra, which could reveal how sensitivity to storage varies by specific freezing temperature
ranges. Bounds on the impact of storage will enable more meaningful intercomparisons of datasets and illuminate best
practices for preserving INPs for offline analysis.
*Data Availability*: The data set supporting this manuscript is hosted by the UCSD Library Digital Collections
(https://doi.org/10.6075/J0M32T8B).
*Supplement Link:* The supplement related to this article is available online at:
*Author Contributions*: CMB wrote the manuscript, prepared figures, led the field campaign and laboratory analysis. DL
contributed to the preparation of figures, precipitation sample collection and laboratory analysis. MDS, TCH, PJD and KAP
provided feedback on the analyses and manuscript. KAP and PJD are principal investigators on awards CHE-1801971 and
AGS-1451347.
*Competing Interests*: The authors declare no competing interests.
*Acknowledgements*: This work was supported by NSF through the NSF Center for Aerosol Impacts on Chemistry of the
Environment (CAICE) CHE- 1801971 and AGS-1451347The National Weather Service Weather Prediction Center provided
the surface analysis satellite composite products used in this study. The AR data were provided by Bin Guan via
https://ucla.box.com/ARcatalog. Development of the AR detection algorithm and databases was supported by NASA.

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

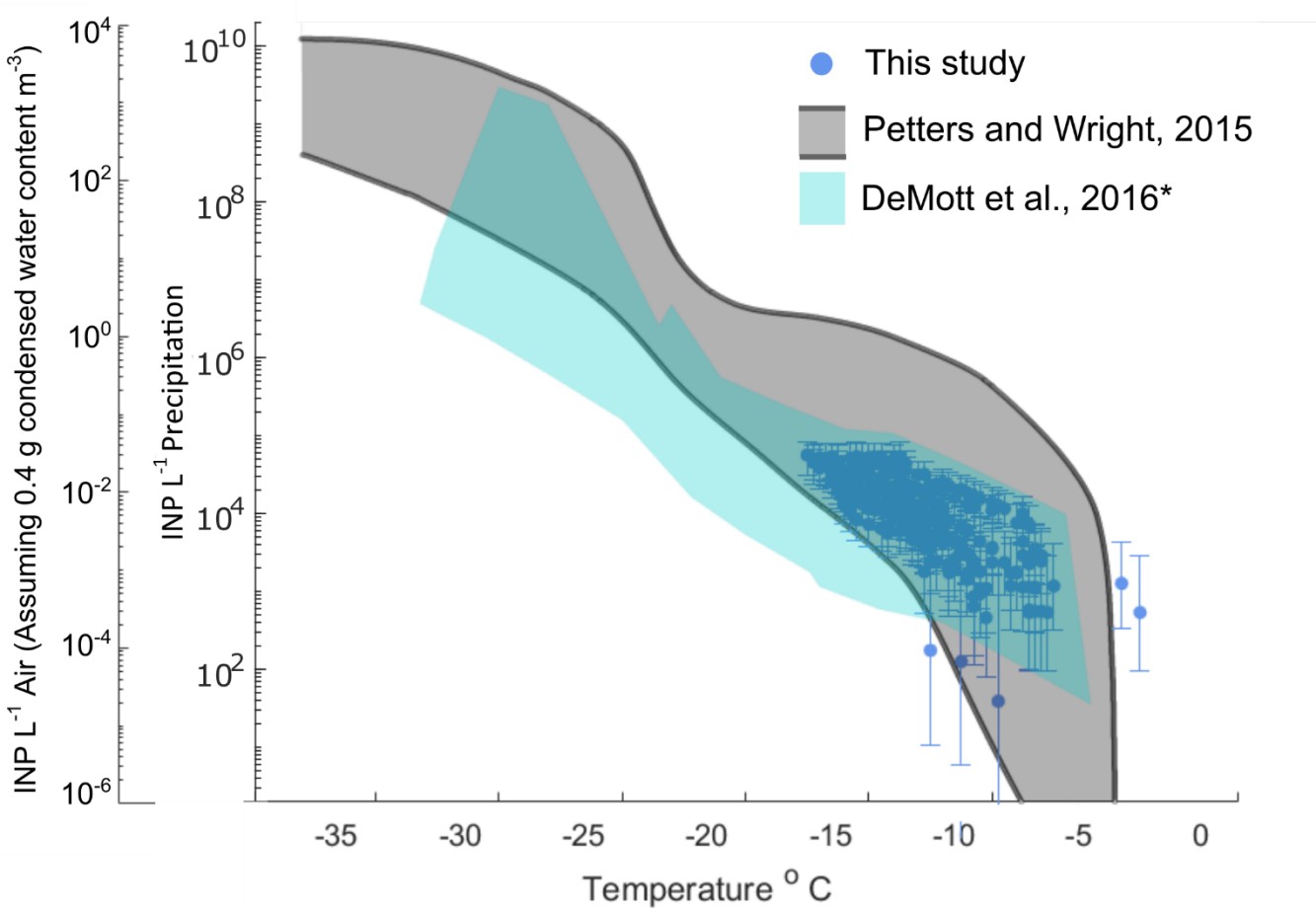

526

**Figure 1: INP concentrations per liter of precipitation and estimated in-cloud INP concentrations per volume of air in 15 precipitation samples collected at two coastal sites at Scripps Institution of Oceanography (La Jolla, California, USA) between 9/22/2016 and 11/22/2019.** Grey shaded region indicates the spectrum of INP concentrations reported in 9 previous studies of precipitation and cloud water samples collected from various seasons and locations worldwide, adapted from Fig. 1 in (Petters and Wright, 2015). The blue shaded region denotes the composite spectrum of INP concentrations observed in a range of marine and coastal environments including the Caribbean, East Pacific and Bering Sea as well as laboratory-generated nascent sea spray (DeMott et al., 2016).

534 *DeMott et al., 2016 data has been updated with a completed dataset for the ICE-T study, as shown in Yang et al., 2020

535

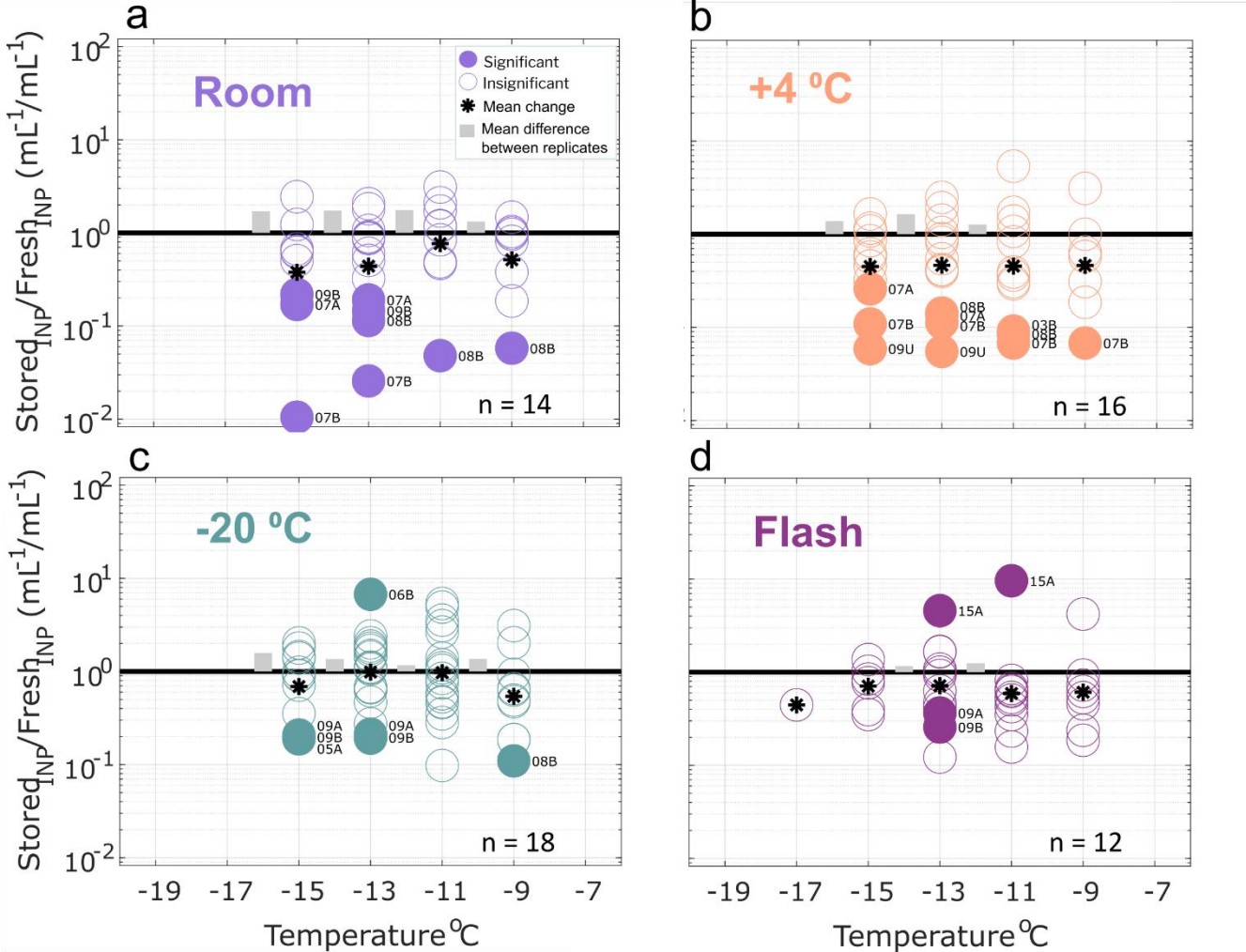

**Figure 2: Ratio of INP concentrations measured in untreated precipitation samples (stored:fresh), calculated in successive 2 °C increments between -19 and -7 °C.** Four storage protocols were applied: **(a)** room temperature (21-23 °C), **(b)** refrigerated (+4 °C), **(c)** frozen (-20 °C) and **(d)** flash freezing in liquid nitrogen before storing frozen (-20 °C). All samples were processed at one or two time intervals between 1 and 166 days post-collection (see Figs S1-S4). For samples processed at two intervals, both replicate samples are represented in the figure for a total of 14, 16, 18 and 12 samples in (a), (b), (c) and (d), respectively (see Table 2 for summary of sample and replicate numbers). Markers above black 1:1 line indicate enhancement of INP concentrations in stored samples, and markers below indicate losses. In temperature intervals containing stored:fresh ratios from at least two sets of replicate samples, grey bars represent the average difference between replicates. Stored sample frozen well fractions that passed Fishers Exact Test ($p < 0.01$) for significant differences from original fresh sample frozen well fractions at each of the 5 temperatures are indicated with filled markers, and the mean change in each temperature interval is marked with a star. Significant data are also labelled to indicate the sample number (01-15, see Table 1), and replicate ("A" or "B", and "U" indicates there were no replicates for the sample). Results show that on average, INP concentrations decrease in stored samples, and that both room temperature storage and refrigeration result in significant INP losses. Frozen and flash frozen storage show comparable results, with fewer (3-4) of the observations exhibiting significant losses and enhancements in INP concentrations.

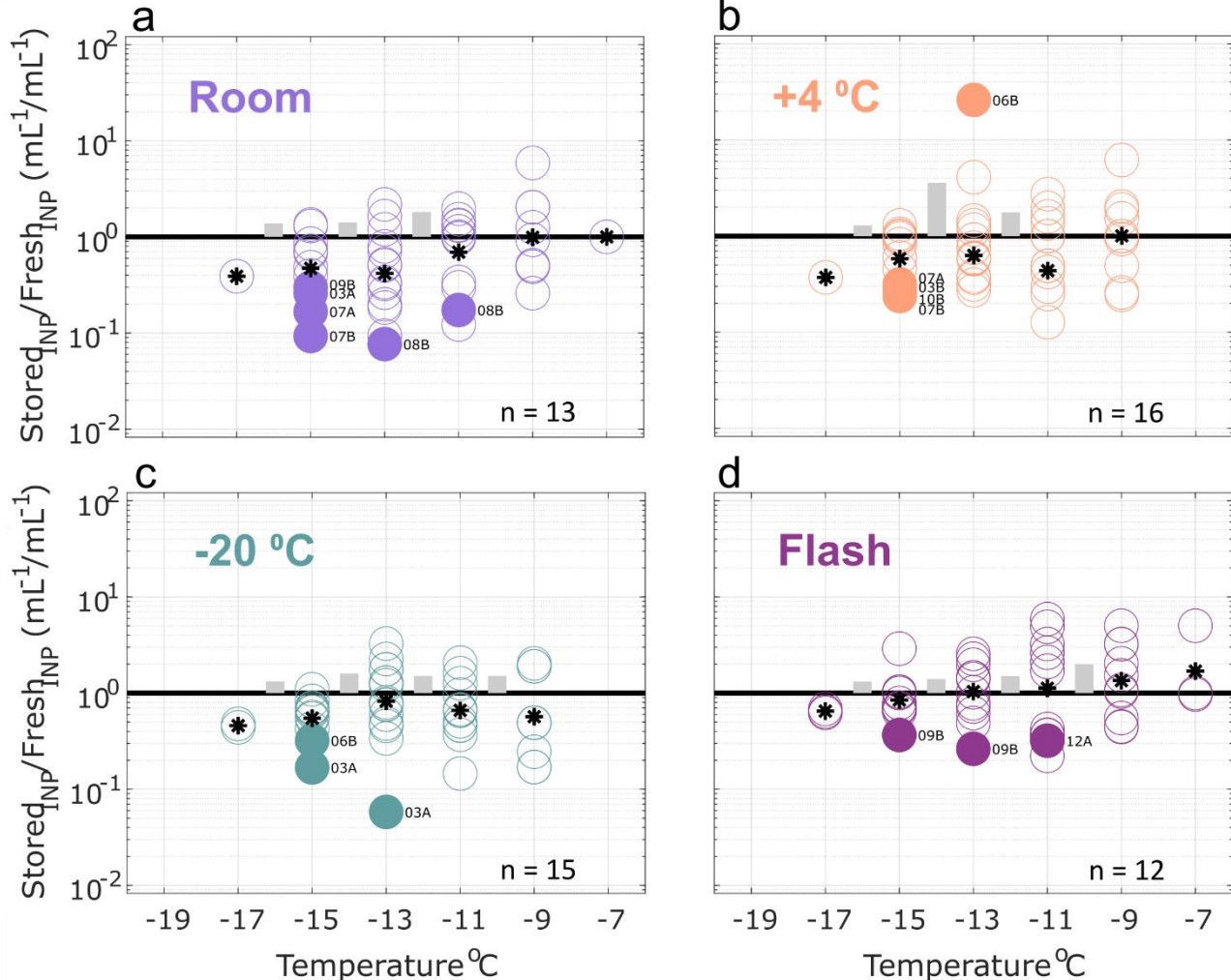

**Figure 3: Ratio of INP concentrations measured in heated precipitation samples (stored:fresh), calculated in successive 2 °C increments between -19 and -7 °C.** Same samples as shown in Figure 2, but heated to 95 °C for 20 minutes prior to measurement to eliminate heat-labile INPs (see Methods Sect. 2.2 for details). All samples were processed at one or two time intervals between 1 and 166 days post-collection. For samples processed at two intervals, both replicate samples are represented in the figure for a total of 13, 16, 15 and 12 samples in (a), (b), (c) and (d), respectively (see Table 3 for summary of sample and replicate numbers). In temperature intervals containing stored:fresh ratios from at least two sets of replicate samples, grey bars represent the average difference between replicates. Results show significant losses of INPs in heat-treated samples stored at room temperature. Refrigerated, frozen, and flash frozen samples show comparable results with a few (1-3) samples exhibiting significant losses and enhancements. Non-heat-labile INPs are generally less sensitive to storage protocol than the total INP population in precipitation samples (Fig. 2), with the exception of storage at room temperature.

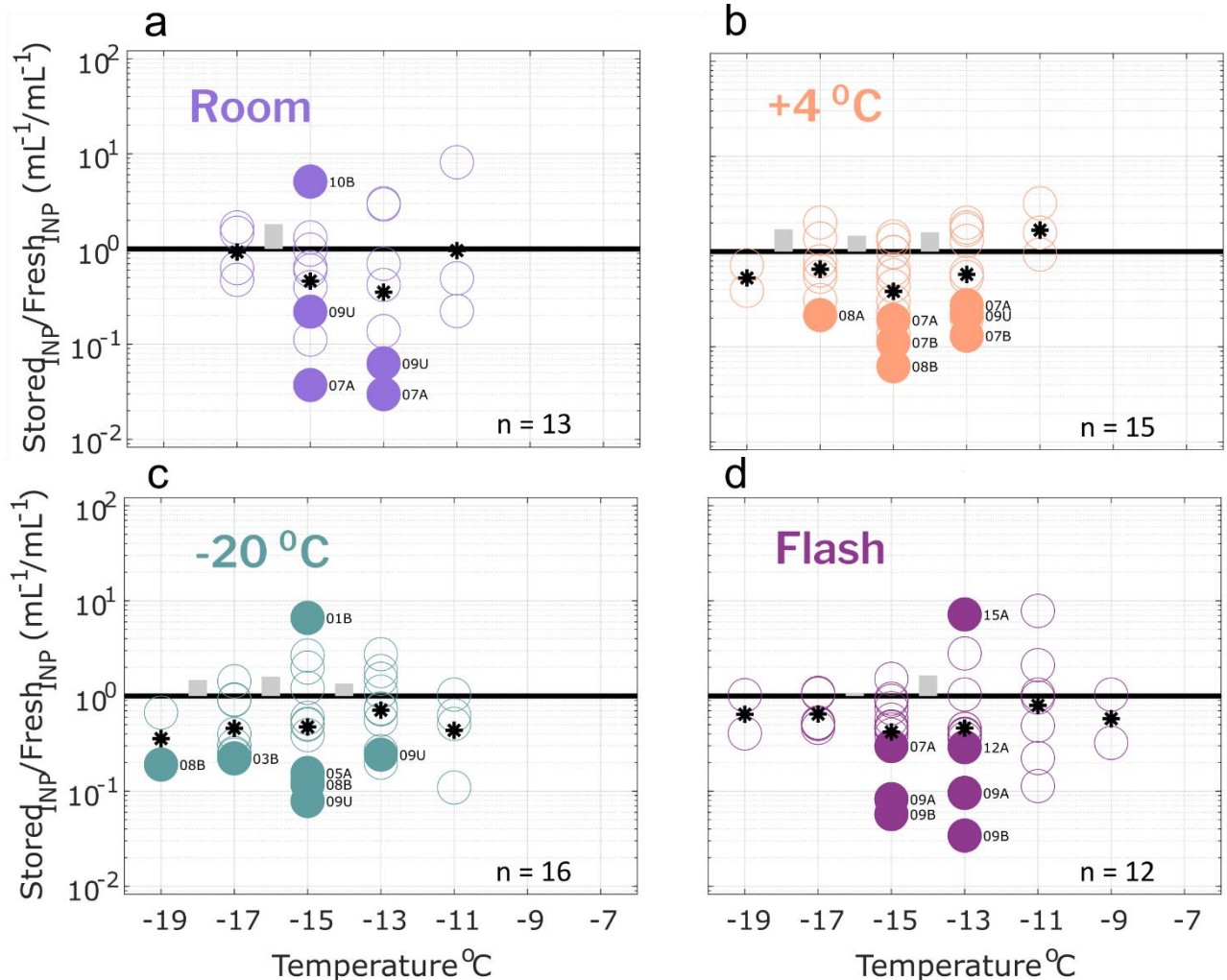

566

Figure 4: Ratio of INP concentrations measured in filtered (0.45 µm) precipitation samples (stored:fresh), calculated in successive 2 °C increments between -19 and -7 °C. Same samples as in Fig. 2 but filtered with a 0.45 µm syringe filter prior to measurement (see Methods Sect. 2.2 for details). All samples were processed at one or two time intervals between 1 and 166 days post-collection. For samples processed at two intervals, both replicate samples are represented in the figure for a total of 13, 15, 16 and 12 samples in (a), (b), (c) and (d), respectively (see Table 4 for summary of sample and replicate numbers). In temperature intervals containing stored:fresh ratios from at least two sets of replicate samples, grey bars represent the average difference between replicates. Results show significant losses of INPs in several filtered samples, regardless of storage protocol.

**Table 1. Precipitation sampling periods**

| Sampling Period | UTC Date | UTC time start | UTC time end | Meteorological Conditions |
|---|---|---|---|---|
| 1 | 9/22/2016 | 19:20 | 21:13 | scattered, low coastal clouds, lack of dynamical system |
| 2 | 9/22/2016 | 19:42 | 21:13 | scattered, low coastal clouds, lack of dynamical system |
| 3 | 12/31/2016 | 4:53 | 7:52 | warm, low cloud rain |
| 4 | 1/1/2017 | 7:53 | 10:52 | post-frontal rain, meso-scale system |
| 5 | 1/5/2017 | 21:02 | 22:01 | pre-frontal rain, meso-scale system |
| 6 | 1/9/2017 | 15:51 | 19:50 | decaying atmospheric river |
| 7 | 1/11/2017 | 19:00 | 23:30 | frontal rain |
| 8 | 1/14/2017 | 2:03 | 6:00 | warm, low cloud rain |
| 9 | 1/19/2017 | 12:30 | 17:30 | pre-frontal rain, meso-scale system |
| 10 | 1/20/2017 | 14:15 | 02:20 (next day) | weak atmospheric river |
| 11 | 11/19/2019 | 22:34 | 22:45 | pre-frontal rain, meso-scale system |
| 12 | 11/22/2019 | 4:43 | 5:42 | scattered, low coastal clouds, lack of dynamical system |
| 13 | 11/22/2019 | 6:43 | 7:42 | scattered, low coastal clouds, lack of dynamical system |
| 14 | 11/23/2019 | 7:42 | 8:41 | convective, local updraft rain |
| 15 | 11/23/2019 | 8:42 | 9:41 | convective, local updraft rain |



**Table 2.  Summary of unique and replicate untreated precipitation samples used for INP concentration measurements featured in Fig. 2.**

| Storage technique | No. of unique samples | No. of stored samples measured at 2 timesteps |
|---|---|---|
| Room temperature (19 - 23 ºC) | 8 | 6 |
| Refrigeration (+4 ºC) | 8 | 8 |
| Freezing (-20 ºC) | 9 | 9 |
| Flash freezing (-20 ºC) | 8 | 4 |

**Table 3.  Summary of unique and replicate heat-treated**
**precipitation samples used for INP concentration**
**measurements featured in Fig. 3.**

| Storage technique | No. of unique samples | No. of stored samples measured at 2 timesteps |
|---|---|---|
| Room temperature (19 - 23 ºC) | 8 | 6 |
| Refrigeration (+4 ºC) | 8 | 8 |
| Freezing (-20 ºC) | 8 | 7 |
| Flash freezing (-20 ºC) | 8 | 4 |


**Table 4.  Summary of unique and replicate filtered (0.45 µm)**
**precipitation samples used for INP concentration**
**measurements featured in Fig. 4.**

| Storage technique | No. of unique samples | No. of stored samples measured at 2 timesteps |
|---|---|---|
| Room temperature (19 - 23 ºC) | 8 | 5 |
| Refrigeration (+4 ºC) | 8 | 7 |
| Freezing (-20  ºC) | 9 | 7 |
| Flash freezing (-20 ºC) | 8 | 4 |


**Table 5. Estimate of uncertainty associated with storage impacts for INPs with activation temperatures between -9 and -17 °C measured in stored, untreated precipitation samples.** Confidence intervals were derived from the log-normal distribution of changes observed in INP concentrations due to storage (see Fig. 2 and details in Sect. 3.2). Temperature intervals where datapoints were too few to derive confidence intervals are indicated with "NA". Changes in INP concentration corresponding to enhancements or losses greater than 1 order of magnitude (losses <= -90% or enhancements >= +900%) in bold.

| Storage protocol | Mean Change -9 °C (%) | 95% CI Low (%) | 95% CI High (%) | Mean Change -11 °C (%) | 95% CI Low (%) | 95% CI High (%) | Mean Change -13 °C (%) | 95% CI Low (%) | 95% CI High (%) | Mean Change -15 °C (%) | 95% CI Low (%) | 95% CI High (%) |
|---|---|---|---|---|---|---|---|---|---|---|---|---|
| Room temperature (21 - 23 ºC)* | -26 | -82 | +200 | -51 | **-97** | +850 | -77 | **-98** | +220 | -77 | **-99** | **+1000** |
| Refrigeration (+4 ºC)* | -42 | -74 | +32 | -74 | **-99** | +400 | -46 | **-95** | +520 | -56 | **-95** | +290 |
| Freezing (-20 ºC) | -48 | **-95** | +430 | -16 | **-90** | +580 | +24 | -80 | +650 | -50 | **-90** | +150 |
| Flash freezing (-20 ºC) | -21 | **-90** | +520 | -41 | **-95** | +560 | -33 | **-91** | +390 | NA | NA | NA |

* For INPs with freezing temperatures >= -9 °C, changes in INP concentrations are moderately correlated with time in samples stored at room temperature or at +4 °C (see Sec. 3.2). Change factors for room temperature and refrigerated storage protocols are derived from samples stored in ranges of 27 – 76 and 8 – 46 days, respectively.

**Table 6. Estimate of uncertainty associated with storage impacts for INPs with activation temperatures between -9 and -17 °C measured in stored, heat-treated precipitation samples.** Confidence intervals were derived from the log-normal distribution of changes observed in INP concentrations due to storage (see Fig. 3 and details in Sect. 3.2). Changes in INP concentration corresponding to enhancements or losses greater than 1 order of magnitude (losses <= -90% or enhancements >= +900%) in bold.

| Storage protocol | Mean Change -9 °C (%) | 95% CI Low (%) | 95% CI High (%) | Mean Change -11 °C (%) | 95% CI Low (%) | 95% CI High (%) | Mean Change -13 °C (%) | 95% CI Low (%) | 95% CI High (%) | Mean Change -15 °C (%) | 95% CI Low (%) | 95% CI High (%) |
|---|---|---|---|---|---|---|---|---|---|---|---|---|
| Room temperature (21 - 23 ºC) | +32 | -74 | +550 | -17 | -86 | +380 | -65 | **-95** | +155 | -58 | **-93** | +150 |
| Refrigeration (+4 ºC) | -56 | **-91** | **+940** | -74 | **-99** | **+1600** | -58 | **-99** | **+6000** | -60 | -87 | +27 |
| Freezing (-20 ºC) | -55 | **-91** | +130 | -53 | -87 | +69 | -42 | **-93** | +390 | -34 | -70 | +47 |
| Flash freezing (-20 ºC) | +36 | -76 | +660 | +31 | -88 | **+1300** | -9.0 | -81 | +340 | +1.0 | -60 | +150 |

**Table 7. Estimate of uncertainty associated with storage impacts for INPs with activation temperatures between -11 and -19 °C measured in stored, filtered precipitation samples.** Confidence intervals were derived from the log-normal distribution of changes observed in INP concentrations due to storage (see Fig. 2 and details in Sect. 3.2). Temperature intervals where datapoints were too few to derive confidence intervals are indicated with "NA". Changes in INP concentration corresponding to enhancements or losses greater than 1 order of magnitude (losses <= -90% or enhancements >= +900%) in bold.

| Storage protocol | Mean Change -11 °C (%) | 95% CI Low (%) | 95% CI High (%) | Mean Change -13 °C (%) | 95% CI Low (%) | 95% CI High (%) | Mean Change -15 °C (%) | 95% CI Low (%) | 95% CI High (%) | Mean Change -17 °C (%) | 95% CI Low (%) | 95% CI High (%) |
|---|---|---|---|---|---|---|---|---|---|---|---|---|
| Room temperature (21 - 23 ºC) | NA | NA | NA | -80 | **-99** | +360 | -72 | **-96** | +130 | -7.0 | -68 | +170 |
| Refrigeration (+4 ºC) | NA | NA | NA | -48 | **-94** | +300 | -65 | **-97** | +250 | -14 | -80 | +250 |
| Freezing (-20 ºC) | NA | NA | NA | -31 | -89 | +330 | -54 | **-98** | +870 | -32 | -78 | +110 |
| Flash freezing (-20 ºC) | -26 | -83 | +230 | -65 | **-98** | +650 | -68 | **-96** | +140 | NA | NA | NA |

608