# Peer review of "Best practices for precipitation sample storage for offline studies of ice nucleation in marine and coastal environments"

_Atmospheric Measurement Techniques, 2020_

## Referee Comment (RC1) · Anonymous Referee #1 · 9 Jul 2020

Beall et al., describes a methodology used by a collaborative group of researchers collecting and then investigating precipitation samples. While the predominant focus is post-collection, laboratory ice nucleation studies it is also a methodology of use for other analytical techniques. The topic of the manuscript is not ground-breaking; storage and subsequent use of precipitation samples has been in place for many decades. However, the description of the methodology is useful for the field and worthy of a publication in AMT.

Beall et al. is a generally well written paper but there are some issues that the authors should consider in revision. Overall, the paper needs to expand the description of the experiments and their context to the larger literature. There are also several statements of cause – effect and/or attribution which, based on the experiments, seem to actually

be theory or assumption by the authors. These should be cleaned up so as to not misinform the reader.

Comments:

1. The 15 samples used as the basis of this paper seems a reasonable number; however, they are all from the same location and therefore likely have a similar mix of INP types. Do the researchers have access to more diverse samples? Would these results hold if the sample was e.g. collected from a boreal forest? A desert? (see also comment on the 'correction factor' in point 3). Given the expertise in the author list, it would seem that access to a variety of samples would be possible. To emphasize this point : the title of this paper is ""Best practices for precipitation sample storage for offline studies of ice nucleation" – of universal importance. But in the Discussion "The aim of this study was to identify a storage protocol . . . in a coastal environment." – much more limited. The 'coastal' modifier is then repeatedly used but this isn't even universally coastal – it is a single coastal location. Either the authors should place the much more geographically restrictive information up front – title and abstract – or provide a larger diversity of samples. The latter, clearly, would be much more beneficial to the field as a whole. 2. Follow on. : At what time of year / conditions were the samples collected? Are these from the same or similar events? What is the diversity of conditions (season, meteorological, etc.)? 3. Starting in the Abstract and continuing through the paper : ". . . non-heat-labile INPs being generally less sensitive to storage regime. . ." "Non-heat-labile INPs were generally less sensitive" This seems to be an assumption; the experiment determines abundances of heat or non-head labile INPs before and after but can not directly say something was changed or not. The authors should indicate that, based on abundances, they assume that the storage process is responsible for the change but not absolutely attribute it. As an example, a constant abundance could mean that no change was caused by storage or that there were roughly equal rates of enhancement and deactivation; the measurements made would not be able to differential this, correct? 4. If the assumption that heat labile INPs

are more sensitive to storage, I don't believe the authors can offer (again, point made in Abstract and continuing through paper):"correction factors are provided so that INP measurements obtained from stored samples may be used to estimate concentrations in fresh samples"– wouldn't said correction factor necessarily be a function of the ratio of non- to heat labile INPs? Therefore the correction factor would not be universal but a function of the INP mix? 5. Introduction "Measurements of INPs suspended in precipitation are commonly made offline using a droplet freezing assay technique, and many studies report results from samples stored prior to processing. Storage protocols vary widely, including total storage time, time between collection and storage, and temperature fluctuations between collection, shipment and storage (if these details are provided at all), yet generally samples are stored between + 4 °C and -20 °C (see Table S1)." – These two sentences follow on a paragraph on INP in clouds. They are disparate concepts and should represent two new paragraphs: (1) how are off-line INP measurements made (they are not only by drop freezing assay – that is only the technique used here)? and (2) there should be a more complete description of storage used by previous researchers, not just a statement that it varies widely / table reference. 6. Discussion, last paragraph starts "Significant enhancements in INP concentrations occurred less frequently than losses. Again, changes in the total particle size distribution could explain some of the observed INP concentration enhancements." – an important conclusion. However, the paragraph then changes topics to the impact of freezing on IN-active (biological) molecules. This is neither consistent with the topic of the paragraph nor is it part of the research outlined in the paper. Lines 259-269, as currently constituted, should be removed.

Grammatical / Spelling

Abstract "…likely and an additional uncertainty in INP concentrations…" remove and?

"Significant insights have been obtained…" 'highly uncertain" : please eliminate non-objective terms like 'significant' (throughout paper) – these are reader dependent, not quantitative.

---

## Referee Comment (RC2) · Gabor Vali (Referee) · 12 Jul 2020

The authors examine a procedural step in the process usually employed for determining the ice nucleating particle (INP) content of precipitation. The experiment they designed is well conceived and the study was thoroughly done. The data obtained is limited in extent and some clarifications are needed about details of the work but the indication for detectable differences among the storage methods seems clear. Importantly, the differences are not of such magnitude as to raise serious questions about results accumulated in past research. The results provide an indication that samples should be stored frozen in future work. The usefulness of the correction factors derived in the paper is doubtful. The authors performed a good experiment; the changes here suggested refer to the analyses of the results and to the clarification of some details of

the procedures.

GENERAL COMMENTS

The paper has two dimensions. One is the application of the results in handling of precipitation samples for analyses of ice nucleating particle (INP) content. The other is the possibility to make inferences about the nature of the INPs in terms of persistence, size and heat resistance. Ultimately, both are parts of the larger question of what can be learned about what is the role of INPs in precipitation formation in the atmosphere, and about what are the sources of those INPs. This paper set out to address a relatively modest part of the larger question but inevitably the results have to be examined in the larger context.

Measurements of INPs in precipitation collected at the ground has a long history (a useful list is found in the Supplement to this paper). These measurements have many caveats attached when making inferences about precipitation formation. The caveats relate to the assumption that INPs found in the precipitation were present at the formation of the cloud and the development of precipitation. Passive scavenging during the fall of already developed precipitation particles is known to contribute to the INP content but, as a first approximation, assumed to be negligible. This view is a reflection of the importance attached to INPs in initiating precipitation in many clouds, and is reinforced by their paucity.

Inferences based on analyses of the INP content of precipitation amount to 'reverse engineering' of the cloud processes. The samples obtained for analyses combine a vary large number of precipitation elements (drops and/or ice crystals) and the INPs are a very minor component of the overall mix of particulate and dissolved impurities. All this is well understood in principle but difficult to quantify.

Immersion-freezing is known from laboratory data to be more effective than other pathways of ice nucleation so drop-freezing assays of various types have gained prominence in this research, specially because INPs active at small supercooling can be

detected. Great care is generally taken in such work to avoid contamination in the process of collection and in the handling of the samples in the laboratory where the INP analyses are performed. The manner of storage of the samples between collection and analysis frequently has been assumed to be inconsequential, for the most part relying on the fact that INPs are insoluble particles. Also, inter-comparison of samples is frequently the goal and identical storage for all samples is assumed to be assurance for the absence of complications. The authors of this paper recognized that the aforementioned assumptions deserve scrutiny, so they designed and executed an experiment to examine what differences may result from four different storage methods. Their results show that storage is best done with the samples kept frozen.

Looking at the magnitude and the variability of the detected influences, my view is that the problem is not of major importance when compared to other uncertainties related to the use of these measurements. Putting it in another way, the results presented provide some assurance that the storage of rain samples between collection and analysis is not a limiting factor in extracting useful results on the INP content of precipitation, and that freezing of the samples is better than storage in liquid state. For snow and hail samples this is obvious, but aging of samples can't be excluded a priori even for those. All the above points are, of course, subject to more tests of storage effects with a greater variety of samples.

The authors raise relevant issues regarding possible influences, specially of freezing and of heat treatment, on INPs of different composition and size. At the moment these are useful speculations that would probably need to be examined with specifically designed experiments using INPs of known sizes and composition. Heat treatment effects are fairly well demonstrated to provide useful distinctions between organic and mineral INPs. Size-dependence of the effect of freezing is a new issue to my knowledge.

The following comments relate to how well the conclusions stated above are supported by data presented in the paper. Overall, the answer is positive, but there are simplifications and gaps that need to be recognized.

MAJOR POINTS

There is no indication in the paper of the kind of precipitation that was sampled. Presumably - vaguely deduced from the variations in the lengths of the sampling periods - a variety of precipitation types are included. Probably, some light rain to more showery situations were involved. Cases with all warm-rain processes and cases with ice origin may have been involved. This may justify the choice of 'precipitation' in the title rather than 'rain'. If all events were from clouds with no ice-phase, a change in the title would be warranted to indicate so. This point isn't very important to the main theme of the paper, but it could possibly make a difference for considerations of how the present results might apply to other situations. The main constraint mentioned in the paper and explicitly stated in the conclusions is that the results refer to coastal environments. This is not as helpful as could be, since precipitation and aerosol sources on the coasts may still include a very broad variety.

Separating measurement variability from actual changes is important. Figures 2-4 include indications of measurement reproducibility with the gray bars adjacent to the data point clusters. All of these bars are indicating values greater than unity. The caption to Fig. 2 says that the bars represent the 'average difference between replicates'. How is this to be interpreted? What conclusion can drawn from these data?

Considering the effects of heat treatment and of filtering in conjunction with storage methods is useful since these are applied in many studies of INP composition and source. A lingering uncertainty in the paper about whether these treatments were applied to the fresh sample before division and storage, or just prior to INP measurement, is disconcerting. The discussion in lines following 218 seem to indicate that filtering was done before freezing for storage. It would be good to have the sequence better described. The overall effects of the treatments are given as, on the average, 59% of INPs were found resistant to heat and 69% passed the filters. These numbers are overly vague, as dependence of temperature can be expected as well as variations from sample to sample. While such detail will not alter the data, it is relevant to possible explanations of the results. On the level of internal consistency in the paper, it is worth asking how justified is the statement underlying conclusion 6 (line 280). Significantly greater losses are said to occur in storage for filtered samples. This is not really evident from a comparison of Fig. 2 with Fig. 4, or from the figures in Table 5 versus Table 7. Greater variability (larger 95% range) is found only for 'refrigeration' and 'freezing', while 'room temperature' and 'flash freezing' have narrower ranges and smaller standard deviations in Table 7 than in Table 5. Perhaps the claimed effect was clear for selected samples but not for the combined sample set.

Tables 5-7 have some technical problems (see comment below on lines 185-189), but taking the data as is, most notable is the large range of variations for the corrections factors. Not just the 95% range, but even 50% spread: for the last line in Table 5, the 50% range is roughly 0.78 to 2.8. Experiments seldom lead to more accurate INP concentrations due to limitations in sample sizes (number of drops or vials). This reinforces the point that the results should be viewed as indications of the uncertainties associated with aging of samples during storage and not as correction factors that can usefully improve measured INP data in other studies. This argument is further supported by the potential for differences in the aging effects for precipitation at different times and locations. The current data provide help in weighing the importance of aging versus other sources of uncertainties in a given experimental design

MINOR POINTS:

How was the sample division done for different treatments? While this can be expected to be a step without risk of introducing discrepancies among the samples, it is not without such a possibility. Thus, the manner it was done should be described, as well as any tests done to assure that this step doesn't lead to artifacts.

Line 85 mentions samples getting divided into 24 bottles during collection. What is the relationship between this and the division of the samples for different treatments? Figure 1 shows points near -5°C for one sample. This should be of special interest but

the paper doesn't mention it. Was the sample unusable?

Perhaps Fig.1 could be made less congested by showing only the interval 0°C to -20°C. Is there more than one point included in Figs. 2-4 for a sample from the same rain event and time period? Unfortunately, one can't determine from the figures how many data points are shown for each temperature. More than the number of rain events? The number of points differs for different temperatures. Is this because of limits in the temperature range of freezing events?

It would have been useful to identify by precipitation sample each data point in Figs 2-4, at least for the outliers. This would clarify, for example, if all the points with lowest values in Fig 2(a) are for the same sample or not, and if the same sample has the lowest points in Fig 2(b) and 2(c).

Line 73: Were heat treatment and filtering applied before division for different storage temperatures, or just before INP analysis? One can assume it is the former, but the paper leaves it unspecified. Lines 101-102 still don't make clear what was done. Line 179 seems to indicate that filtration was done before storage.

Line 100: reference to 'section above' seems incorrect

Line 171: Is the ratio cited independent of the INP activity temperature?

Line 138: ' ... binned by 2°C increments .. ' seems odd for cumulative data. More likely, values are 'determined (or calculated) at successive 2°C intervals'. If that is not the case, please explain. The word 'binned' appears in numerous places in the text.

Line 140: What does 'significant' refer to here? Maybe the authors meant 'measured'.

Line 141: The cumulative values at any point are calculated by accounting for all freezing events (all frozen sample wells) at temperatures higher than the value at which the concentrations is evaluated, not just those of the preceding value at 2°C higher temperature. Also, in line 146, 'each' should be replaced by 'all', and line 147 should be rephrased and clarified.

Line 157: ' .. containing data from at least two sets of replicate samples ...' seems to say that data points shown include replicates from the same rain event. This is brought up again in lines 187-188 and in the caption to Fig. 2.

Lines 160-161: 'well counts' and 'well fractions' are not the same - please clarify.

Line 161: ' ... at each of the 5 temperatures ..' should probably be left out

Line 163: Here it says that all stored samples showed significant changes whereas only a few points are shaded in Fig. 2.

Line 179: The reference to Sect. 2.3 for detail is incorrect.

Lines 185-189: Tables 5-7 indicate the range of impacts that may be expected on the basis of the data presented in this paper. The correction factors here given appear to have been derived combining data from all temperatures for given storage and treatment type. This has an inherent multiplicity problem as data at successively lower temperatures include all data from higher temperatures. Thus, a value for, say, -11°C is also incorporated into the values at -13°C, -15°C etc. so that the ratio at -11°C is given multiple, though uneven, weights when combining all the values for -11°C and lower into calculating a mean and standard deviation for the given treatment. Also, all data were included in calculating the values in Tables 5-7, not just the cases for which the differences observed were shown to be statistically significant. One may wonder what the results would be of only those cases were included.

Line 192: What is meant by 'in situ' collection? Similarly, in line 241 'in situ dust' is vague.

---

## Referee Comment (RC3) · Anonymous Referee #4 · 10 Aug 2020

**General comments:**

This manuscript provides a systematic study of the effect of storage conditions on the number and activity of ice-nucleating particles (INP) in precipitation samples. The study has implications for numerous past and future studies of INPs, and it fits the scope of Atmospheric Measurement Techniques as it provides "techniques of data processing and information retrieval for [...] aerosols [...]".

The paper text, length, and most of the figures are appropriate. However, I have some specific reservations regarding some of the analysis and related figures. For example, figures S1-S4 should be revised and the related correlation analysis, too. Moreover, I do not agree with the recommendation of specific correction factors. The strength of the paper is not the provision of specific correction factors for various storage protocols

[Figure]

– in my mind the usefulness of such factors are questionable – but by showing which storage protocols are most suitable for offline measurements of INPs, and by making suggestions for sensitivity analyses that should go along with every such measurement in the future. For more details, see the specific comments below.

In summary, this is a useful paper, which may be publishable in Atmospheric Measurement Techniques after the comments below have been considered.

**Scientific comments:**

(1) Figure 1 and line 136: I am wondering, why the error bars shown are symmetrical. In a log-plot, I would assume unsymmetrical bars for symmetrical errors.

(2) L.138: Section 3.2: I am missing storage experiments with 'pure' water, since we know from our own experiments that even deionized/distilled water can become ice-nucleating after several days.

(3) L.192-193, L262-263 (conclusion), L.292-293: Apparently, there are significant deviations in the stored:fresh ratios, both above and below 1. How can then simple correction factors be applied? In addition, I am highly skeptical about these correction factors: given that the actual correction factors are usually small (mostly between 0.9 and 1.8), they are likely much smaller than most other errors in such type of ice nucleation studies, and so their usefulness is questionable in my view. In addition, it is highly questionable whether these correction factors can be applied to studies at other locations, using different sampling and investigation methods, and studying different (marine) samples. I would very much prefer to see instead the uncorrected raw data then in such studies. In summary, I do not concur with conclusion no. 2. The authors also seem to be skeptical as they state in lines 292-293: "However, it remains to be seen how INP sensitivity to storage varies by environment or INP composition."

(4) Figure S1-S4: I do not understand what is plotted in Figures S1 through S4 in

the supplement, and I am in doubt that it is correct. The captions say "INP losses or enhancements (%) ..." What are losses in %? Shouldn't they be given as negative numbers? How can losses and enhancements be fitted simultaneously as a basis for correlation analysis, as the figure captions imply?

Even if not losses in percent are meant but if loss factors are presented, then losses would imply values smaller than 1. However, in none of the figures S1-S4 is there any point below the 10^0 line. How can that be, as figures 2-4 of the main paper clearly show that losses do occur?

Moreover, I am wondering whether plotting the losses or enhancements in percent does make sense at all. I think factors would be more suitable, because some of the changes are several orders of magnitudes. In particular for losses (not such much for enhancements), plotting them in percent may be misleading: for example losses by a factor of 10^-2 or 10^-4 (i.e., a difference of two orders of magnitude) would lead to a loss of nearly -100% in both cases (-99% or -99.99%). Note that losses cannot be lower than -100%!

**Minor and technical comments:**

(5) L.88: "At the MESOM Laboratory parking lot..." To which of the two collection points given (lines 81-83) does this location belong?

(6) L.262-263: "... it is worth noting that freezing is lethal for most cells" This statement is too general. Note that INTRACELLULAR freezing is lethal for most cells, while EXTRACELLULAR freezing is often not critical and, thus, survived by freeze-tolerating species.

(7) L.458 (caption to Fig.3): "measured in heated precipitation samples" When were the samples heated? Directly after collection, or just before measurement?

(8) L.468 (caption to Fig.4): "measured in filtered (0.45 $\mu$m) precipitation samples"

When were the samples filtered? Directly after collection, or just before measurement?

(9) Tables 5-7: Please provide a few sentences of explanation on the 95% confidence interval limits. What exactly do these values imply and, more importantly, how can they be applied? For example, considering line 2 in Table 5: the suggested correction factor is 1.72. The confidence limits of this correction factor are 0.25 and 11.27, implying that the correction factor could also be significantly below 1. I was wondering then, given this large confidence interval, whether it is useful at all to make such a correction (see also my comment 3 above).

---

## Author Comment (AC1) · 7 Sep 2020

The Authors wish to thank Referee 1 for their comments and discussion. We summarize our responses and changes made to the manuscript below. Note that line numbers refer to the latest, marked-up version of the manuscript:

Referee: "1. The 15 samples used as the basis of this paper seems a reasonable number; how-ever, they are all from the same location and therefore likely have a similar mix of INP types. Do the researchers have access to more diverse samples? Would these results hold if the sample was e.g. collected from a boreal forest? A desert? (see also comment on the 'correction factor' in point 3). Given the expertise in the author list, it would seem that access to a variety of samples would be possible.

[Figure]

To emphasize this point : the title of this paper is ""Best practices for precipitation sample storage for offline studies of ice nucleation" – of universal importance. But in the Discussion "The aim of this study was to identify a storage protocol...in a coastal environment."– much more limited. The 'coastal' modifier is then repeatedly used but this isn't even universally coastal – it is a single coastal location. Either the authors should place the much more geographically restrictive information up front – title and abstract – or provide a larger diversity of samples. The latter, clearly, would be much more beneficial to the field as a whole."

We agree that a study with a greater diversity of samples and sites could be more broadly applicable, and perhaps provide greater insights. A focus on one location for demonstrating impacts of storage on precipitation samples allowed for a manageable study as one part of a PhD, and serves the purpose of highlighting that the lack of an existing standard storage protocol in the field is potentially problematic. These findings, thus, will serve as motivation for future efforts to quantify storage impacts on samples from a variety of environments, which will require either a series of field campaigns or a coordinated study between groups at different institutions/locations. To provide broader context for our dataset, we have updated Figure 1 following the assumptions of Petters and Wright, 2015 to estimate in-cloud INP concentrations from precipitation samples (i.e. 0.4 g condensed water content m-3). Figure 1 shows that the INPs observed here are comparable to spectra reported previously for a wide range of marine and coastal environments, including the Caribbean, Bering Sea, East Pacific and nascent sea spray aerosol (DeMott et al., 2016). As INP spectra in this temperature regime cluster distinctly by air mass type (e.g. Figure 1-10 in Kanji et al., 2017), Fig. 1 indicates that the air masses sampled in this study were likely primarily marine. Regarding the reviewer's point "[...] this isn't even universally coastal [. . .]", we agree that it remains to be seen how sensitivity to storage varies between sites with similar source types. This reinforces the need for future studies of the effects of storage, not only upon INPs in precipitation, but also with filter or impinger samples that are used in investigations globally.

We have made the following changes to explicitly state this assumption and reflect it consistently throughout the text:

Title: "Best practices for precipitation sample storage for offline studies of ice nucleation in marine and coastal environments"

Abstract, Line 25: "We provide the following recommendations for preservation of precipitation samples from coastal or marine environments intended for INP analysis..."

Introduction, Line 100: "Enhancements and losses of INPs according to storage protocol and treatment are reported, as well as recommendations for storage protocols that best preserve INPs in untreated, heated, and filtered precipitation samples from marine or coastal environments."

Sec 3.1, Line 171: " Following the assumptions in (Wright and Petters, 2015) to estimate in-cloud INP concentrations from precipitation samples (i.e. condensed water content of 0.4 g m-3 air), observations of INP concentrations in fresh precipitation samples are additionally compared to studies of field measurements conducted in marine and coastal environments. Figure 1 shows that atmospheric INP concentration estimates compare with INP concentrations observed in a range of marine and coastal environments, including the Caribbean, East Pacific, and Bering Sea, as well as laboratory-generated nascent sea spray aerosol (DeMott et al., 2016)."

Discussion, Line 282: "Additionally, the INP freezing temperatures and concentrations observed in this study compare with INPs observed in studies of marine and coastal environments (Fig. 1). As spectra in this regime (-5 to -20 °C and 10-5 to ~10-1 per L air, respectively) cluster distinctly by source type (see Fig. 1-10 in Kanji et al., 2017), Fig. 1 indicates that the dominant sources to air masses sampled in this study were marine. Considering that data in this study compare well with marine and coastal INPs from a variety of marine-influenced air masses (DeMott et al., 2016, Yang et al., 2019), the findings herein are likely relevant to samples from other marine and coastal environments."

Discussion, Line 297: "If correspondence within 1 order of magnitude (or 2-3 °C) is desired, uncertainties associated with storage should also be considered in studies using samples from coastal or marine environments. Thus, uncertainty distributions provided in Tables 5-7 can be used to evaluate observed INP concentrations and responses to treatments in the context of potential changes due to storage. However, the degree to which INP sensitivity to storage varies by INP source (e.g. with soil-derived INP populations) remains to be tested."

Conclusions, Line 370: "Based on all observations in this study, we provide the following recommendations for precipitation samples collected in coastal and marine environments for offline INP analyses..."

Referee: "2. Follow on. : At what time of year / conditions were the samples collected? Are these from the same or similar events? What is the diversity of conditions (season, meteorological, etc.)"

A summary of the meteorological conditions associated with each sample have been added to Table 1, and the following changes have been made to the text: Sec. 2.1, Line 117: "Satellite composites from the National Weather Service Weather Prediction Center's North American Surface Analysis Products were used for synoptic weather analysis to generally characterize each rain event (see Table 1). Atmospheric river (AR) events were identified using the AR Reanalysis Database described in (Guan and Waliser, 2015) and (Guan et al., 2018)."

Sec. 3.1 Line 165: "Figure 1 shows INP concentrations of 15 coastal rain samples, collected in a variety of meteorological conditions including scattered, low coastal rainclouds, frontal rain, and atmospheric river events (see Table 1)."

Referee: "3. Starting in the Abstract and continuing through the paper : "...non-heat-labile INPs being generally less sensitive to storage regime..." "Non-heat-labile INPs were generally less sensitive" This seems to be an assumption; the experiment determines abundances of heat or non-head labile INPs before and after but can not

directly say something was changed or not. The authors should indicate that, based on abundances, they assume that the storage process is responsible for the change but not absolutely attribute it. As an example, a constant abundance could mean that no change was caused by storage or that there were roughly equal rates of enhancement and deactivation; the measurements made would not be able to differential this, correct?"

The referee makes a good point that there are other potential causes of changes in INPs that should be discussed. Sample handling procedures, for example, could cause apparent differences in INP concentrations, or contamination in storage containers. However, we are able to distinguish changes due to sample handling and changes due to storage by considering the differences between sample replicates. The following changes to the text have been made:

Sec 3.2, line 218: "Replicate samples were processed for each storage protocol so that impacts of sample handling can be distinguished from storage impacts. For example, if settling occurs in bulk rain samples that are then divided into smaller volumes prior to storage, INP concentrations may differ between replicates of the bulk sample. . Thus, it is assumed that INP concentration changes that are greater than differences between replicates (grey bars in Figs 2-4) can be attributed to storage impacts. We also assume that stored:fresh INP concentration ratios of 1:1 indicate insensitivity to storage, although it is possible that enhancements and losses of equal magnitude could also result in a 1:1 concentration ratio."

Referee: If the assumption that heat labile INPs are more sensitive to storage, I don't believe the authors can offer (again, point made in Abstract and continuing through paper):"correction factors are provided so that INP measurements obtained from stored samples may be used to estimate concentrations in fresh samples"– wouldn't said correction factor necessarily be a function of the ratio of non- to heat labile INPs? Therefore the correction factor would not be universal but a function of the INP mix?

Due to referee #2's point that many of the correction factors are within measurement uncertainty for droplet assay techniques, we have updated Tables 5-7 with average changes in INP concentration and 95% confidence intervals that can be used to estimate uncertainty associated with storage in samples from marine and coastal environments (see response to point #1 above). Emissions of heat-labile particles can be increased in bloom-enhanced conditions, although it is variable (McCluskey et al., 2018), and considering bloom timescales (e.g. 10 days), bloom-enhanced marine sources would not have dominated air-masses sampled in a precipitation study.

The following changes have been made to the text to reflect this update:

[revised manuscript text omitted]

Referee: "6. Discussion, last paragraph starts "Significant enhancements in INP concentrations occurred less frequently than losses. Again, changes in the total particle size distribution could explain some of the observed INP concentration enhancements." – an important conclusion. However, the paragraph then changes topics to the impact of freezing on IN-active (biological) molecules. This is neither consistent with the topic of the paragraph nor is it part of the research outlined in the paper. Lines 259-269, as currently constituted, should be removed."

These lines have been removed from the last paragraph of the Discussion.

Abstract ": : :likely and an additional uncertainty in INP concentrations: : :" remove and?

Corrected. Abstract, Line 25: "We provide the following recommendations for preservation of precipitation samples from coastal or marine environments intended for INP analysis: that samples be stored at -20 °C to minimize storage artifacts, that changes due to storage are likely an additional uncertainty in INP concentrations..."

"Significant insights have been obtained: : :" 'highly uncertain" : please eliminate nonobjective terms like 'significant' (throughout paper) – these are reader dependent, not quantitative.

These terms have been removed from the paper, except for instances referring to Fisher's Exact Test. A statement to clarify this has been added to the Results section 3.2.

Sec. 3.2, Line 229: "The term "significant" henceforth is intended to describe INP losses or enhancements that correspond to frozen well fractions that are determined to be significantly different from corresponding fresh sample frozen well fractions, according to Fisher's Exact Test (i.e. filled markers in Figs. 2-4)."

Introduction, Line 78: "The understanding of storage effects on INPs suspended in precipitation is limited (Petters and Wright, 2015)..."
* * *
[Figure]

[Figure]

**Fig. 1.** Figure1

---

## Author Comment (AC2) · 7 Sep 2020

The Authors would like to thank Gabor Vali for his helpful suggestions and discussion. A summary of our responses is below. Line numbers refer to the latest marked-up version of the manuscript.

Referee: "There is no indication in the paper of the kind of precipitation that was sampled. Presumably - vaguely deduced from the variations in the lengths of the sampling periods - a variety of precipitation types are included. Probably, some light rain to more showery situations were involved. Cases with all warm-rain processes and cases with ice origin may have been involved. This may justify the choice of 'precipitation' in the title rather than 'rain'. If all events were from clouds with no ice-phase, a change in

the title would be warranted to indicate so. This point isn't very important to the main theme of the paper, but it could possibly make a difference for considerations of how the present results might apply to other situations."

Thank you for this suggestion. A summary of the types of precipitation events that were sampled has been added to Table 1. Meteorological conditions associated with precipitation ranged from AR events to warm, low altitude rain clouds. It is true that the choice of "precipitation" in the title and throughout the text was motivated by ambiguity regarding the ice or liquid origin of the samples. Although all samples were collected as liquid at ground-level, it is possible that ice-processes were involved in the precipitating clouds (judging by the low cloud-top temperatures in the NWS satellite composite analysis).

The following updates have been made to the text: Sec. 2.1, Line 117: Satellite composites from the National Weather Service Weather Prediction Center's North American Surface Analysis Products were used for synoptic weather analysis to generally characterize each rain event (see Table 1). Atmospheric river (AR) events were identified using the AR Reanalysis Database described in (Guan and Waliser, 2015) and (Guan et al., 2018).

Sec. 3.1, Line 165: Figure 1 shows INP concentrations of 15 coastal rain samples, collected in a variety of meteorological conditions including scattered, low coastal rain-clouds, frontal rain, and atmospheric river events (see Table 1). Observations generally fall within bounds of previously reported INP concentrations from precipitation and cloud water samples (grey shaded region, adapted from Petters and Wright, 2015). Observed freezing temperatures ranged from -4.0 to -18.4 °C, with concentrations up to the limit of testing at 105 INP L-1 precipitation. AIS measurement uncertainties are represented with 95% binomial sampling intervals (Agresti and Coull, 1998).

Referee: "The main constraint mentioned in the paper and explicitly stated in the conclusions is that the results refer to coastal environments. This is not as helpful as could

be, since precipitation and aerosol sources on the coasts may still include a very broad variety."

To provide broader context for our dataset, we have updated Figure 1 following the assumptions of Petters and Wright, 2015 to estimate in-cloud INP concentrations from precipitation samples (i.e. 0.4 g condensed water content m-3). The updated Figure 1 shows that the INPs observed here are comparable to spectra reported previously for a wide range of marine and coastal environments, including the Caribbean, Bering Sea, East Pacific and nascent sea spray aerosol (DeMott et al., 2016). As INP spectra in this temperature regime cluster distinctly by air mass type (e.g. Figure 1-10 in Kanji et al., 2017), Fig. 1 indicates that the air masses sampled in this study were likely primarily marine.

The following updates have been made to the text:

Title: "Best practices for precipitation sample storage for offline studies of ice nucleation in marine and coastal environments"

Sec. 3.1, Line 171: " Following the assumptions in (Wright and Petters, 2015) to estimate in-cloud INP concentrations from precipitation samples (i.e. condensed water content of 0.4 g m-3 air), observations of INP concentrations in fresh precipitation samples are additionally compared to studies of field measurements conducted in marine and coastal environments. Figure 1 shows that atmospheric INP concentration estimates compare with INP concentrations observed in a range of marine and coastal environments, including the Caribbean, East Pacific, and Bering Sea, as well as laboratory-generated nascent sea spray aerosol (DeMott et al., 2016)."

Sec. 3.1, Line 282: "Additionally, the INPs in this study compare with INPs observed in studies of marine and coastal environments (Fig. 1). As spectra in this regime (-5 to -20 °C and 10-5 to ~10-1 per L air, respectively) cluster distinctly by source type (see Fig. 1-10 in Kanji et al., 2017), Fig. 1 indicates that the dominant sources to air masses sampled in this study were marine. Considering that data in this study

are characteristic of marine and coastal INPs previously reported over a wide range of marine environments (DeMott et al., 2016), we assume that the findings herein are relevant to samples from other marine and coastal environments."

Sec. 3.1: Line 294: If correspondence within 1 order of magnitude (or 2-3 °C) is desired, uncertainties associated with storage should also be considered in studies using samples from coastal or marine environments. Thus, uncertainty distributions provided in Tables 5-7 can be used to evaluate observed INP concentrations and responses to treatments in the context of potential changes due to storage. However, the degree to which INP sensitivity to storage varies by INP source (e.g. with soil-derived INP populations) remains to be tested.

Referee: "Separating measurement variability from actual changes is important. Figures 2-4 include indications of measurement reproducibility with the gray bars adjacent to the data point clusters. All of these bars are indicating values greater than unity. The caption to Fig. 2 says that the bars represent the 'average difference between replicates'. How is this to be interpreted? What conclusion can drawn from these data?"

The following changes have been made in Sec 3.2 to describe how grey bars are to be interpreted and conclusions that may be drawn from the data. This section has also been updated to address the referee's 1st comment on sample handling in "Minor points":

Sec. 3.2, Line 218: Replicate samples were processed for each storage protocol so that impacts of sample handling can be distinguished from storage impacts. For example, if settling occurs in bulk rain samples that are then divided into smaller volumes prior to storage, INP concentrations may differ between replicates of the bulk sample. Thus, it is assumed that INP concentration changes that are greater than differences between replicates (grey bars in Figs 2-4) can be attributed to storage impacts.

Referee: "A lingering uncertainty in the paper about whether these treatments were applied to the fresh sample before division and storage, or just prior to INP measurement, is disconcerting. The discussion in lines following 218 seem to indicate that filtering was done before freezing for storage. It would be good to have the sequence better described."

Thank you for bringing this to our attention. The following change has been made to Sec. 2.2 Storage Protocols:

Line 136: Heat treatments and filters were applied to samples just prior to processing (i.e. treatments were not applied to samples prior to storage).

Referee: "The overall effects of the treatments are given as, on the average, 59% of INPs were found resistant to heat and 69% passed the filters. These numbers are overly vague, as dependence of temperature can be expected as well as variations from sample to sample. While such detail will not alter the data, it is relevant to possible explanations of the results."

This is a good point. The figures quoted above were calculated at the temperature of the next to last freezing event of the corresponding fresh sample (beyond which the data is not meaningful), again using the cumulative distribution. I recalculated the ratios of heat-treated and filtered to untreated INPs in fresh samples in the temperature intervals consistent with the rest of the manuscript (-9, -11, -13, -15 etc). This new way of calculating the filtered to untreated fractions yields a different answer regarding the general sizes of the INPs, interestingly, probably because smaller particles represent larger fraction of INPs only at the colder temperatures. It is also now necessary to discuss that we observed some enhancements in INPs after heating fresh samples (5 of the 15 samples). We have added the following detail and discussion:

Sec 3.1, Line 178: "In 5 of the 15 heat-treated samples, INP concentrations were increased by 1.9 – 13X between -9 and -11 °C (see Discussion). Excluding these 5 samples, the fraction of heat-resilient INPs varied between samples and generally increased with decreasing temperature. Geometric means and standard deviations of heat-treated:untreated INP ratios were 0.40 ×/Ãů 1.9, 0.51 ×/Ãů 2.0, and 0.62 ×/Ãů

2.1 at -11 , -13, and -15 °C respectively. Fractions of INPs < 0. 45 $\mu$m also varied between samples, with geometric means and standard deviations of 0.48 ×/Ãů 1.73, 0.30 ×/Ãů 3.4 and 0.37 ×/Ãů 1.9 at -11 , -13, and -15 °C respectively. Mean values of heat-resilient INP fractions and INPs < 0.45 $\mu$m were calculated using the geometric mean, which is more appropriate than the arithmetic mean for describing a distribution of ratios (Fleming and Wallace, 1986)."

Discussion, Line 272: "The fractions of INPs < 0.45 $\mu$m observed in this study varied between 52 and 63% at -11 and -15 °C, respectively. Excluding the five heat-treated samples in which INP concentrations were enhanced (e.g. 1.9 - 13X between -9 and -11 °C), the average fraction of non-heat-labile INPs varied between 40 and 62% at -11 and -15 °C, respectively. INP enhancements in heat-treated samples are unexpected, as heat-treatments are typically applied assuming that heat destroys proteinaceous (e.g. biological) INPs. The causes of INP enhancements in heat-treated samples are unknown and have only been reported in coastal precipitation samples (Martin et al., 2017) and nascent sea spray aerosol (McCluskey et al., 2018).

Discussion, Line 305: "Despite the range of enhancements and losses of heat-sensitive INPs observed in fresh samples, non-heat-labile INPs were generally less sensitive to storage than the total INP population., and with the exception of samples stored at room temperature, all techniques yielded similar results with fewer enhancements or losses. "

Discussion, Line 330: In this study, a large fraction (30% to 48%, on average) of INPs observed in fresh precipitation samples were < 0.45 $\mu$m. Considering this and that INPs < 0.45 $\mu$m exhibit significant losses across all storage types, there is a risk that filter-treatments on stored samples in this study would lead to the underestimation of INPs < 0.45 $\mu$m.

Referee: On the level of internal consistency in the paper, it is worth asking how justified is the statement underlying conclusion 6 (line 280). Significantly greater losses

are said to occur in storage for filtered samples. This is not really evident from a comparison of Fig. 2 with Fig. 4, or from the figures in Table 5 versus Table 7. Greater variability (larger 95% range) is found only for 'refrigeration' and 'freezing', while 'room temperature' and 'flash freezing' have narrower ranges and smaller standard deviations in Table 7 than in Table 5. Perhaps the claimed effect was clear for selected samples but not for the combined sample set.

This conclusion is based on the increased frequency of significant (Fisher's, p<0.01) data points on Figs 2 and 4. After making the suggested changes (see response to comment below), the 95% confidence intervals span losses > 1 order of magnitude across all protocols and most temperature intervals. We have added the following text to explain how Tables 5-7 may be interpreted:

Discussion, Line 291: While mean INP changes are within a factor of ∼2 or less of fresh sample INP concentrations for all protocols except "Room temperature" (Table 5), none of the 4 storage protocols prevented significant losses or enhancements of INP concentrations in all samples (Fig. 2), indicating that INP concentration measurements on fresh precipitation are superior to measurements on stored samples. 95% confidence intervals in Table 5 span losses > 1 order of magnitude in all protocols across multiple temperature intervals. As uncertainties < 1 order of magnitude are necessary for the quantitative comparison between studies (DeMott et al., 2017), our results demonstrate that uncertainties associated with storage must be considered in studies of stored samples from coastal or marine environments. Thus, the uncertainty distributions provided in Tables 5-7 can be used to evaluate observed INP concentrations and responses to treatments in the context of potential changes due to storage. However, the degree to which INP sensitivity to storage varies by INP source (e.g. with soil-derived INP populations) remains to be tested.

Referee: "Tables 5-7 have some technical problems (see comment below on lines 185-189), but taking the data as is, most notable is the large range of variations for the corrections factors. Not just the 95% range, but even 50% spread: for the last line

in Table 5, the 50% range is roughly 0.78 to 2.8. Experiments seldom lead to more accurate INP concentrations due to limitations in sample sizes (number of drops or vials). This reinforces the point that the results should be viewed as indications of the uncertainties associated with aging of samples during storage and not as correction factors that can usefully improve measured INP data in other studies. This argument is further supported by the potential for differences in the aging effects for precipitation at different times and locations. The current data provide help in weighing the importance of aging versus other sources of uncertainties in a given experimental design."

We agree that these results are better indications of the uncertainties associated with storage. Updates have been made to the tables, table legends and text to reflect this change:

Abstract: Finally, the estimated uncertainties associated with the 4 storage protocols are provided for untreated, heat-treated and filtered samples for INPs between -9 and -17 °C. Conclusion, Line 375: "2. Estimates of uncertainty attributed to storage impacts and 95% confidence intervals for INP measurements obtained from stored samples are provided (see Tables 5-7)."

Legend Table 5: Table 5. Estimate of uncertainty associated with storage impacts for INPs with activation temperatures between -9 and -17 °C measured in stored, untreated precipitation samples. Confidence intervals were derived from the log-normal distribution of changes observed in INP concentrations due to storage (see Fig. 2 and details in Sect. 3.2). Temperature intervals where datapoints were too few to derive confidence intervals are indicated with "NA". Changes in INP concentration corresponding to enhancements or losses greater than 1 order of magnitude (losses <= -90% or enhancements >= +900%) in bold.

Legend Table 6: Table 6. Estimate of uncertainty associated with storage impacts for INPs with activation temperatures between -9 and -17 °C measured in stored, heat-treated precipitation samples. Confidence intervals were derived from the log-normal

distribution of changes observed in INP concentrations due to storage (see Fig. 3 and details in Sect. 3.2). Changes in INP concentration corresponding to enhancements or losses greater than 1 order of magnitude (losses <= -90% or enhancements >= +900%) in bold.

Legend Table 7: Table 7. Estimate of uncertainty associated with storage impacts for INPs with activation temperatures between -11 and -19 °C measured in stored, untreated precipitation samples. Confidence intervals were derived from the log-normal distribution of changes observed in INP concentrations due to storage (see Fig. 2 and details in Sect. 3.2). Temperature intervals where datapoints were too few to derive confidence intervals are indicated with "NA". Changes in INP concentration corresponding to enhancements or losses greater than 1 order of magnitude (losses <= -90% or enhancements >= +900%) in bold.

Minor points: Referee: "How was the sample division done for different treatments? While this can be expected to be a step without risk of introducing discrepancies among the samples, it is not without such a possibility. Thus, the manner it was done should be described, as well as any tests done to assure that this step doesn't lead to artifacts."

The following updates have been made to the text: Sec. 2.2, Line 128: Prior to storage, 25 - 50 mL bulk sample aliquots were distributed directly from collection bottles into Falcon® tubes, shaking bottles ~10 s between each distribution.

Sec. 3.2, Line 218: Replicate samples were processed for each storage protocol so that impacts of sample handling can be distinguished from storage impacts. For example, if settling occurs in bulk rain samples that are then divided into smaller volumes prior to storage, INP concentrations may differ between replicates of the bulk sample. Thus, it is assumed that INP concentration changes that are greater than differences between replicates (grey bars in Figs 2-4) can be attributed to storage impacts.

Referee: "Line 85 mentions samples getting divided into 24 bottles during collection. What is the relationship between this and the division of the samples for different treatments?"

This is a description of the precipitation collection device, which changes bottles on a rotating carousel at the specified time interval. There is no consistent relationship between the bottle numbers and sample division because sometimes we combined bottles corresponding to consecutive 1-hour sampling intervals in order to have enough volume for each of the sampling protocols, replicates, treatments, etc. This is described in Sec. 2.1: "The samples were distributed via the distributor arm into one of twenty-four 1-liter polypropylene bottles on an hourly time interval. Bottles were combined when the hourly precipitation volume was insufficient for sample separation and analysis (< 50 mL)."

I have updated the last line to help clarify: Sec. 2.1, Line 112: "Bottles corresponding to consecutive 1-hour time intervals were combined when the hourly precipitation volume was insufficient for sample separation and analysis (< 50 mL per bottle)."

Referee: "Figure 1 shows points near -5◦C for one sample. This should be of special interest but the paper doesn't mention it. Was the sample unusable?"

We have added the following text to acknowledge the two warm-freezing observations (this sample was used in the storage experiments). Sec. 3.1, Line 176: "However, two of the warmest-freezing INP observations in Fig. 1 (at -4.0 and -4.75 °C) exceed temperatures commonly observed in marine-influenced atmospheres, precipitation and cloudwater samples

Referee: "Perhaps Fig.1 could be made less congested by showing only the interval 0◦C to -20◦C." This is true, however, I think that seeing the whole Wright and Petters, 2015 and now DeMott et al., 2016 composite spectrum because it provides context for the regime we are observing.

Referee: "Is there more than one point included in Figs. 2-4 for a sample from the same rain event and time period? Unfortunately, one can't determine from the figures

how many data points are shown for each temperature. More than the number of rain events? The number of points differs for different temperatures. Is this because of limits in the temperature range of freezing events?"

Yes, more than one point from the same sample is included in Figs. 2-4. This was motivated by the fact that a subset of the replicate samples exhibited differing sensitivity to storage (Fig. S5). However, replicates were not included in uncertainty factor calculations to avoid underweighting the small subset of samples that did not have replicates due to sample volume limitations. This is currently stated in the figure legends and table legends, and tables 2-4 show how many unique samples are represented in each figure vs how many replicates. The number of points differ due to limits of detection. There are typically fewer datapoints in the warmest and coldest temperature bin. At the warmest temperature bin, one of the samples (fresh or stored) is more likely to have 0 wells frozen, which would result in either a ratio of 0 or Inf. Ratios of zero were excluded because they are reflective of the limit of detection due to the number of droplets processed rather than a true lack of ice nucleating particles at this temperature. Similarly, datapoints tend to be fewer in the coldest temperature bin because in one or both of the samples (fresh and stored), all the wells had frozen.

The following has been added to the text: Sec. 3.2 Line 191: "Numbers of datapoints in Figs 2-4 differ across the temperature intervals due to limits of detection (i.e. ratios were not calculated at temperatures where zero or all wells were frozen in the fresh and/or stored sample)."

Figure 2 legend: All samples were processed at one or two time intervals between 1 and 166 days post-collection (see Figs S1-S4). For samples processed at two intervals, both replicate samples are represented in the figure for a total of 14, 16, 18 and 12 samples in (a), (b), (c) and (d), respectively (see Table 2 for summary of sample and replicate numbers). Significant data are also labelled to indicate the sample number (01-15, see Table 1), and replicate ("A" or "B", and "U" indicates there were no replicates for the sample). Figure 3 legend: All samples were processed at one or two

time intervals between 1 and 166 days post-collection. For samples processed at two intervals, both replicate samples are represented in the figure for a total of 13, 16, 15 and 12 samples in (a), (b), (c) and (d), respectively (see Table 3 for summary of sample and replicate numbers).

Figure 4 legend: All samples were processed at one or two time intervals between 1 and 166 days post-collection. For samples processed at two intervals, both replicate samples are represented in the figure for a total of 13, 15, 16 and 12 samples in (a), (b), (c) and (d), respectively (see Table 4 for summary of sample and replicate numbers).

Figures 2-4 have been also been updated to show the numbers of samples represented for each protocol. Each filled data marker is now also labeled to show which sample and replicate it corresponds to.

Referee: "It would have been useful to identify by precipitation sample each data point in Figs 2-4, at least for the outliers. This would clarify, for example, if all the points with lowest values in Fig 2(a) are for the same sample or not, and if the same sample has the lowest points in Fig 2(b) and 2(c)"

This is a good idea, thank you. Figs 2-4 have been updated as suggested, with some text on each filled marker with the precipitation sample number and either the letter "A" or "B" to indicate which replicate it is. Samples without replicates are marked "U".

Referee: "Line 73: Were heat treatment and filtering applied before division for different storage temperatures, or just before INP analysis? One can assume it is the former, but the paper leaves it unspecified. Lines 101-102 still don't make clear what was done. Line 179 seems to indicate that filtration was done before storage."

Heat treatments and filtering was applied just before INP analysis, since this order is probably more common in e.g. field campaigns where samples must be frozen onsite before transport.

The following was added to Sec. 2.2, Line 136: "Heat treatments and filters were

applied to samples just prior to processing (i.e. treatments were not applied to samples prior to storage)."

Referee: "Line 100: reference to 'section above' seems incorrect."

This line has been corrected in Sec 2.2, Line 133: "INP measurements were made in two or three time steps: within two hours of collection, and once or twice after storing using one of four storage protocols described above, depending on volume."

Referee: "Line 171: Is the ratio cited independent of the INP activity temperature?"

This ratio has been updated to reflect temperature dependence (see responses to referee comment beginning "The overall effects of the treatments are given as, on the average [. . .]")

Referee: "Line 138: ' ... binned by 2◦C increments .. ' seems odd for cumulative data. More likely, values are 'determined (or calculated) at successive 2◦C intervals'. If that is not the case, please explain. The word 'binned' appears in numerous places in the text."

Thank you for bringing this to our attention. The following corrections have been made: Sec. 3.2, Line 189: INP concentrations of stored replicate samples are compared with original fresh precipitation samples in Figures 2-4, calculated in successive 2 °C increments between -7 and -19 °C.

Sec 3.2, Line 194: All stored:fresh ratios were calculated from cumulative INP distributions in 2 °C intervals, meaning that the INP concentration in each interval is inclusive of the concentration in the preceding (warmer) temperature interval. Thus, in this study, deviations observed in a stored sample are not necessarily independent, i.e. the sensitivity of INPs to storage in one temperature interval could impact the observed changes in each of the following (colder) temperature interval. For example, in the fresh untreated precipitation samples (see Fig. 1), the contribution of INPs from the preceding 2 °C interval ranges from 32 to 46% between -9 and -17 °C.

Sec 3.2., Line 216: For each temperature interval containing data from at least two sets of replicate samples, the average difference in stored:fresh concentration ratios between replicates are represented with grey bars to indicate measurement variability.

Sec 3.2, Line 226: Finally, Fisher's Exact Test was applied to frozen and unfrozen well fractions between each stored sample and its corresponding fresh sample at each of the 2 °C temperature intervals.

Sec 3.2, Line 232: Results in Fig. 2 show that significant enhancements or losses of INPs occurred for all stored samples between -9 and -17 °C, and that on average, stored samples exhibit INP losses (as indicated by the mean change in each temperature interval)." Figure 2 legend: Figure 2: Ratio of INP concentrations measured in untreated precipitation samples (stored:fresh), calculated in successive 2 °C increments between -19 and -7 °C. In temperature intervals containing stored:fresh ratios from at least two sets of replicate samples, grey bars represent the average difference between replicates. Stored sample frozen well fractions that passed Fishers Exact Test ($p < 0.01$) for significant differences from original fresh sample frozen well fractions at each of the 5 temperatures are indicated with filled markers, and the mean change in each temperature interval is marked with a star.

Figure 3 legend: Figure 3: Ratio of INP concentrations measured in heated precipitation samples (stored:fresh), calculated in successive 2 °C increments between -19 and -7 °C. In temperature intervals containing stored:fresh ratios from at least two sets of replicate samples, grey bars represent the average difference between replicates Figure 4 legend: Ratio of INP concentrations measured in filtered (0.45 $\mu$m) precipitation samples (stored:fresh), calculated in successive 2 °C increments between -19 and -7 °C. In temperature intervals containing stored:fresh ratios from at least two sets of replicate samples, grey bars represent the average difference between replicates.

Referee: "Line 140: What does 'significant' refer to here? Maybe the authors meant 'measured'."

This line has been corrected in Sec 3.2, Line 190: This temperature range was chosen for the analysis because most fresh precipitation samples exhibited freezing activity between -7 and -19 °C.

Referee: "Line 141: The cumulative values at any point are calculated by accounting for all freezing events (all frozen sample wells) at temperatures higher than the value at which the concentrations is evaluated, not just those of the preceding value at 2°C higher temperature. Also, in line 146, 'each' should be replaced by 'all', and line 147 should be rephrased and clarified."

This line has been corrected in Sec 3.2, Line 194: All stored:fresh ratios were calculated from cumulative INP distributions in 2 °C intervals, meaning that the INP concentration in each interval is inclusive of the concentration in all of the preceding (warmer) temperature intervals.

Line 201: For example, in fresh untreated precipitation samples (see Fig. 1), 32% of the INP concentration calculated at -11 °C activated in one of the preceding (warmer) 2 °C temperature intervals. At -17 °C, this fraction is increased to 46%.

Referee: "Line 157: ' .. containing data from at least two sets of replicate samples ...' seems to say that data points shown include replicates from the same rain event. This is brought up again in lines 187-188 and in the caption to Fig. 2."

Yes, this line is intended to make clear that replicates are represented in Fig. 2-4. See response to comment beginning "Is there more than one point included in Figs. 2-4 for a sample from the same rain event and time period?"

Referee: "Lines 160-161: 'well counts' and 'well fractions' are not the same - please clarify."

This line has been corrected in Sec 3.2, Line 229: Finally, Fisher's Exact Test was applied to frozen and unfrozen well fractions between each stored sample and its corresponding fresh sample at each of the 2 °C temperature intervals. Stored sample frozen

well fractions that were significantly different (p < 0.01) from fresh sample frozen well fractions at each of the 5 temperatures are indicated with filled markers.

Referee: "Line 161: ' ... at each of the 5 temperatures ..' should probably be left out"

Line 228 has been corrected: "Stored sample frozen well fractions that were significantly different (p < 0.01) from fresh sample frozen well fractions are indicated with filled markers."

Referee: "Line 163: Here it says that all stored samples showed significant changes whereas only a few points are shaded in Fig. 2"

Line 232 in Sec 3.2. has been corrected: "Results in Fig. 2 show that significant enhancements or losses of INPs occurred in all storage protocols between -9 and -17 °C, and that on average, stored samples exhibit INP losses (as indicated by the mean change in each temperature interval)."

Referee: "Line 179: The reference to Sect. 2.3 for detail is incorrect"

Line 248 in Sec 3.2 has been corrected: Effects of storage protocol on INP concentrations of filtered precipitation samples are shown in Figure 4 (0.45 $\mu$m syringe filter, see Sect. 2.2 for details).

Referee: Lines 185-189: Tables 5-7 indicate the range of impacts that may be expected on the basis of the data presented in this paper. The correction factors here given appear to have been derived combining data from all temperatures for given storage and treatment type. This has an inherent multiplicity problem as data at successively lower temperatures include all data from higher temperatures. Thus, a value for, say, -11°C is also incorporated into the values at -13°C, -15°C etc. so that the ratio at -11°C is given multiple, though uneven, weights when combining all the values for -11°C and lower into calculating a mean and standard deviation for the given treatment. Also, all data were included in calculating the values in Tables 5-7, not just the cases for which the differences observed were shown to be statistically significant.

One may wonder what the results would be of only those cases were included."

Thank you for bringing this to our attention. We agree that there is a multiplicity problem in combining the data from all intervals and have updated the tables to show average changes and confidence intervals for each 2 °C temperature increment. See also responses to p.4 comment beginning "Tables 5-7 have some technical problems [...]" about how the tables and text have been changed to reflect that these data represent uncertainty associated with storage rather than correction factors.

I also agree that it would be interesting to recalculate these figures using the significant (Fisher's) datapoints only, but I couldn't think of a way to justify the exclusion of the insignificant datapoints in a calculation. It is also likely that such a distribution would not be log-normal, and then it wouldn't be meaningful to calculate the geometric mean and 95% confidence intervals.

Referee: "Line 192: What is meant by 'in situ' collection? Similarly, in line 241 'in situ dust' is vague."

Line 263 and Line 338 have been updated in the Discussion section:

The challenge in selecting a storage protocol for atmospheric samples (e.g. precipitation, cloud water, ambient atmosphere) is that the INP population composition is unknown, diverse, and the impact of any given technique on the different species may vary.

Considering that well-characterized IN-active dust and biological standards (Arizona Test Dust and Snomax®, respectively) are sensitive to storage conditions, it is possible that dust or biological INPs contributed to the observed INP losses.

[Figure]

**Fig. 1.** Figure 2

[Figure]

**Fig. 2.** Figure 3

[Figure]

**Fig. 3.** Figure 4

---

## Author Comment (AC3) · 7 Sep 2020

The Authors wish to thank Referee 4 for their comments and discussion. We summarize our responses and changes made to the manuscript below. Note that line numbers refer to the latest, marked-up version of the manuscript:

Referee: "(1) Figure 1 and line 136: I am wondering, why the error bars shown are symmetrical. In a log-plot, I would assume unsymmetrical bars for symmetrical errors."

I am assuming that the referee is referring to the error bars on the most visible points (e.g. the points at -4 and -4.75  $^{\circ}$ C). The error bars represent 95% binomial sampling intervals as in (Agresti and Coull, 1998), and are not symmetrical. I realize the figure is a bit congested and it isn't possible to see the individual data points. I considered

decreasing the x-axis, but given the updates to Fig. 1, I think it is important to keep the full composite INP spectrum visible.

Referee: "(2) L.138: Section 3.2: I am missing storage experiments with 'pure' water, since we know from our own experiments that even deionized/distilled water can become ice nucleating after several days."

This is very interesting, but we have not experienced this ourselves, in DI water that we have stored for weeks in polypropylene bottles, and are not aware of any literature reporting such an effect in deionized, distilled or otherwise 'pure' water. For those reasons we did not carry out such a test.

Referee: "(3) L.192-193, L262-263 (conclusion), L.292-293: Apparently, there are significant deviations in the stored:fresh ratios, both above and below 1. How can then simple correction factors be applied? In addition, I am highly skeptical about these correction factors: given that the actual correction factors are usually small (mostly between 0.9 and 1.8), they are likely much smaller than most other errors in such type of ice nucleation studies, and so their usefulness is questionable in my view. In addition, it is highly questionable whether these correction factors can be applied to studies at other locations, using different sampling and investigation methods, and studying different(marine) samples. I would very much prefer to see instead the uncorrected raw data then in such studies. In summary, I do not concur with conclusion no. 2. The authors also seem to be skeptical as they state in lines 292-293: "However, it remains to be seen how INP sensitivity to storage varies by environment or INP composition.""

Considering these issues and suggestions made by the other reviewers, we agree that the results are better represented as indications of the uncertainties associated with storage. Updates have been made to the tables, table legends and text to reflect this change:

Abstract: Finally, the estimated uncertainties associated with the 4 storage protocols are provided for untreated, heat-treated and filtered samples for INPs between -9 and
-17 °C.

Conclusion, Line 375: "2. Estimates of uncertainty attributed to storage impacts and 95% confidence intervals for INP measurements obtained from stored samples are provided (see Tables 5-7)."

Legend Table 5: Table 5. Estimate of uncertainty associated with storage impacts for INPs with activation temperatures between -9 and -17 °C measured in stored, untreated precipitation samples. Confidence intervals were derived from the log-normal distribution of changes observed in INP concentrations due to storage (see Fig. 2 and details in Sect. 3.2). Temperature intervals where datapoints were too few to derive confidence intervals are indicated with "NA". Changes in INP concentration corresponding to enhancements or losses greater than 1 order of magnitude (losses <= -90% or enhancements >= +900%) in bold.

Legend Table 6: Table 6. Estimate of uncertainty associated with storage impacts for INPs with activation temperatures between -9 and -17 °C measured in stored, heat-treated precipitation samples. Confidence intervals were derived from the log-normal distribution of changes observed in INP concentrations due to storage (see Fig. 3 and details in Sect. 3.2). Changes in INP concentration corresponding to enhancements or losses greater than 1 order of magnitude (losses <= -90% or enhancements >= +900%) in bold.

Legend Table 7: Table 7. Estimate of uncertainty associated with storage impacts for INPs with activation temperatures between -11 and -19 °C measured in stored, filtered precipitation samples. Confidence intervals were derived from the log-normal distribution of changes observed in INP concentrations due to storage (see Fig. 2 and details in Sect. 3.2). Temperature intervals where datapoints were too few to derive confidence intervals are indicated with "NA". Changes in INP concentration corresponding to enhancements or losses greater than 1 order of magnitude (losses <= -90% or enhancements >= +900%) in bold.
Referee: "Figure S1-S4: I do not understand what is plotted in Figures S1 through S4 in the supplement, and I am in doubt that it is correct. The captions say "INP losses or enhancements (%) : : :" What are losses in %? Shouldn't they be given as negative numbers? How can losses and enhancements be fitted simultaneously as a basis for correlation analysis, as the figure captions imply? Even if not losses in percent are meant but if loss factors are presented, then losses would imply values smaller than 1. However, in none of the figures S1-S4 is there any point below the 10ËĘ0 line. How can that be, as figures 2-4 of the main paper clearly show that losses do occur? Moreover, I am wondering whether plotting the losses or enhancements in percent does make sense at all. I think factors would be more suitable, because some of the changes are several orders of magnitudes. In particular for losses (not such much for enhancements), plotting them in percent may be misleading: for example losses by a factor of 10ËĘ-2 or 10ËĘ-4 (i.e., a difference of two orders of magnitude) would lead to a loss of nearly -100% in both cases (-99% or -99.99%). Note that losses cannot be lower than -100%!"

Thank you for bringing this to our attention. Figures S1-S4 were originally intended to show whether absolute change in INP concentration relates to the storage time interval, so that we could determine whether the magnitude of the change correlated in time, independent of the sign of the change. However, the referee brings up a good point that there is a problem of scale as losses approach (-) 100%, but enhancements have no upper bound.

Figures S1-S4 have been updated as suggested using the INP change factors, and the text has been updated as follows (to reflect the updated correlation factors):

Abstract, Line 23: Correlations between total storage time (1-166 days) and changes in INP concentrations were weak across sampling protocols, with the exception of INPs with freezing temperatures >= -9 °C in samples stored at room temperature. Sec. 3.2, Line 209: For INPs with freezing temperatures >= -9 °C in samples stored at room temperature, time is moderately correlated with changes in INP concentrations (R2 =

**AMTD**
0.58).

Conclusions, Line 379: 4. With the exception of warm-freezing INPs (freezing temperatures >=  $-9 \,^{\circ}$ C) in samples stored at room temperature, we found little to no correlation between changes in INP concentrations and storage intervals on timescales between 1-166 days, indicating that most enhancements or losses are likely happening during freezing or on timescales

tion samples (stored:fresh), calculated in successive 2  $^{\circ}$ C increments between -19 and -7  $^{\circ}$ C. Same samples as shown in Figure 2, but heated to 95  $^{\circ}$ C for 20 minutes just prior to measurement to eliminate heat-labile INPs (see Methods Sect. 2.2 for details).

Referee: "(8) L.468 (caption to Fig.4): "measured in filtered (0.45  $\mu$ m) precipitation samples" When were the samples filtered? Directly after collection, or just before measurement?"

This legend has also been updated:

Figure 4 Legend: Figure 4: Ratio of INP concentrations measured in filtered (0.45  $\mu$ m) precipitation samples (stored:fresh), calculated in successive 2 °C increments between -19 and -7 °C. Same samples as in Fig. 2 but filtered with a 0.45  $\mu$ m syringe filter prior to measurement (see Methods Sect. 2.2 for details).

Referee: "(9) Tables 5-7: Please provide a few sentences of explanation on the 95% confidence interval limits. What exactly do these values imply and, more importantly, how can they be applied? For example, considering line 2 in Table 5: the suggested correction factor is 1.72. The confidence limits of this correction factor are 0.25 and 11.27, implying that the correction factor could also be significantly below 1. I was wondering then, given this large confidence interval, whether it is useful at all to make such a correction (see also my comment 3 above)"

Tables 5-7 and text have been updated to reflect changes suggested in the referee's comment #3. Additionally, the following has been added to the text to explain how these values may be interpreted:

Discussion, Line 289: While mean INP changes are within a factor of  $\sim$ 2 or less of fresh sample INP concentrations for all protocols except "Room temperature" (Table 5), none of the 4 storage protocols prevented significant losses or enhancements of INP concentrations in all samples (Fig. 2), indicating that INP concentration measurements on fresh precipitation are superior to measurements on stored samples. 95% confi-

**AMTD**
dence intervals in Table 5 span losses > 1 order of magnitude in all protocols across multiple temperature intervals. These uncertainties equal or exceed INP measurement uncertainties (1-2 orders of magnitude) at temperatures > -20 °C due to discrepancies between instruments (DeMott et al., 2017). If correspondence within 1 order of magnitude (or 2-3 °C) is desired, uncertainties associated with storage should also be considered in studies using samples from coastal or marine environments. Thus, uncertainty distributions provided in Tables 5-7 can be used to evaluate observed INP concentrations and responses to treatments in the context of potential changes due to storage. However, the degree to which INP sensitivity to storage varies by INP source (e.g. with soil-derived INP populations) remains to be tested.

**AMTD**
Fig. 1. Figure S1